# One-Step Flow Policy Mirror Descent

## Abstract

Diffusion policies have achieved great success in online reinforcement learning (RL) due to their strong expressive capacity. However, the inference of diffusion policy models relies on a slow iterative sampling process, which limits their responsiveness. To overcome this limitation, we propose *Flow Policy Mirror Descent (FPMD)*, an online RL algorithm that enables 1-step sampling during flow policy inference. Our approach exploits a theoretical connection between the distribution variance and the discretization error of single-step sampling in straight interpolation flow matching models, and requires no extra distillation or consistency training. We present two algorithm variants based on flow policy and Mean-Flow policy, respectively. Extensive empirical evaluations on MuJoCo and visual DeepMind Control Suite benchmarks demonstrate that our algorithms show strong performance comparable to diffusion policy baselines while requiring orders of magnitude less computational cost during inference.

## 1 Introduction

Diffusion models have established themselves as the state-of-the-art paradigm in generative modeling (Ho et al., 2020; Dhariwal & Nichol, 2021), capable of synthesizing data of unparalleled quality and diversity across various modalities, including images, audio, and video. The success is rooted in a principled, thermodynamically-inspired framework that learns to reverse a gradual noising process (Sohl-Dickstein et al., 2015). Diffusion policies are now being used to create highly flexible and expressive policies for decision-making tasks like robotic manipulation by modeling complex, multi-modal action distributions from demonstration (Chi et al., 2023; Ke et al., 2024; Scheikl et al., 2024). This approach has shown significant promise in both imitation learning and reinforcement learning settings (Wang et al.; Chen et al., 2022; Team et al., 2024), enabling agents to learn flexible and effective behaviors.

Despite the benefits of expressiveness, diffusion policies in online RL suffer from steep computational price for policy inference: the sampling process requires repeated neural-network evaluations to produce a single sample, slowing down both the training and the testing of online RL. This drawback hinders the application of diffusion models in tasks that require real-time and compute-constrained inference, which is critical to many real-world applications, such as motion planning and control. The current diffusion policy for online RL mostly focuses on the efficiency and optimality of the training optimization (Psenka et al., 2023; Ding et al., 2024; Wang et al., 2024; Ren et al., 2024; Ma et al., 2025; Celik et al., 2025), while little attention has been paid to the efficiency of policy inference, which typically relies on more than 10 denoising steps. Although there is recent work incorporating one-step policies, they learn the one-step policy by distilling a multi-step policy (Park et al., 2025; Prasad et al., 2024) or applying additional consistency loss (Ding & Jin, 2023), which is redundant and even induces extra computational cost.

To handle this inference-time problem, we leverage flow-based models (Lipman et al., 2022; Liu et al., 2022; Geng et al., 2025) as the policy structure of online RL. One intriguing property of flow-based models with straight interpolation is that, when the target distribution has zero variance, sampling trajectories are straight lines pointing directly toward the target point (Hu et al., 2024). Moreover, the single-step sampling error is bounded by the variance of the target distribution, allowing one-step generation of flow policy and easier training of MeanFlow policy when the target distribution has a small variance.

However, applying the flow model in online RL is nontrivial. There is no closed-form of $\log$ probability of flow models, making the classic policy gradient non-compatible with flow policy. Fortunately, we exploit the variational technique to derive an equivalent loss for online flow polices, bypassing the inaccessibility of the probability of flow models. We also observe that the progress to train a one-step flow model fits perfectly in the exploration-exploitation trade-off of online RL. During the intermediate stage favoring exploration, the flow parametrization of policy can model highly complex and multi-modal distributions, enabling rich exploratory behaviors. As learning progresses and the policy converges, the optimal distribution favoring exploitation typically exhibits low variance. This low-variance regime aligns with the straight-line interpolation property of flow models, enabling efficient single-step sampling with no additional cost. This provides a free lunch to leverage the rich expressiveness to improve performance without increasing the inference-time computational cost like diffusion models.

Building upon this, we propose *Flow Policy Mirror Descent (FPMD)*, an online RL algorithm that enables one-step sampling during policy inference. We further introduce two variants of FPMD using the flow parametrization and MeanFlow parametrization, denoted FPMD-R and FPMD-M respectively. We conduct extensive evaluations on both state-based Gym MuJoCo environments and visual DMControl environments. Empirical results show that the proposed algorithm achieve performance comparable to diffusion policy baselines while using orders of magnitude less computational cost.

Our core contributions are summarized as follows:

- We propose tractable loss functions to train Flow and MeanFlow policies in online RL, allowing one-step action generation during inference time.

- By making moderate assumptions on the variance of the optimal policy, we theoretically analyze the single-step sampling error of the flow policy.

- We conduct extensive empirical evaluations on Gym MuJoCo and visual DMControl tasks. The proposed algorithm shows strong performance comparable to diffusion policy baselines while requiring orders of magnitude fewer function evaluations.

## 1.1 RELATED WORK

We briefly review the most relevant prior work here and provide the full details in Appendix A.

Diffusion policies have been leveraged in many recent online RL studies due to their expressiveness and flexibility. To obtain tractable training objectives, existing methods explored reparameterized policy gradient (Wang et al., 2024; Ren et al., 2024), weighted self-improvement (Ma et al., 2025; Ding et al., 2024), and other variants of score matching (Psenka et al., 2023; Yang et al., 2023). However, these methods have not considered the inference-time difficulties of diffusion policies. A few recent studies on offline RL (Ding & Jin, 2023; Park et al., 2025) took a step toward one-step action generation but their solutions are complicated and involve multi-stage training. Several concurrent works have explored reinforcement learning for flow policy via reparameterized policy gradient (Lv et al., 2025; Koirala & Fleming, 2025), reward-weighted regression (Pfrommer et al., 2025), and policy gradient with $\log$-likelihood approximation (McAllister et al., 2025). Compared to these methods, ours is the only method that achieves an effective balance between policy distribution expressiveness and action sampling efficiency, by introducing a practical training objective equivalent to the flow matching objective and enabling one-step action generation.

## 2 PRELIMINARIES

**Markov Decision Processes (MDPs).** We consider Markov decision process (Puterman, 2014) specified by a tuple $\mathcal{M} = (\mathcal{S}, \mathcal{A}, r, P, \mu_0, \gamma)$, where $\mathcal{S}$ is the state space, $\mathcal{A}$ is the action space, $r : \mathcal{S} \times \mathcal{A} \to \mathbb{R}$ is a reward function, $P : \mathcal{S} \times \mathcal{A} \to \Delta(\mathcal{S})$ is the transition operator with $\Delta(\mathcal{S})$ as the family of distributions over $\mathcal{S}, \mu_0 \in \Delta(\mathcal{S})$ is the initial distribution and $\gamma \in (0, 1)$ is the discount factor. The goal of reinforcement learning is to find an optimal policy $\pi(\cdot|s) : \mathcal{S} \to \Delta(\mathcal{A})$, which maximizes the discounted cumulative rewards, *i.e.*, $\rho(\pi) := \mathbb{E}\left[\sum_{t=0}^{\infty} \gamma^t r(s_t, a_t)\right]$. Given policy $\pi$, the $Q$-function is defined as

$$Q^\pi(s, a) = \mathbb{E}_{s_{t+1} \sim P(\cdot|s_t, a_t), a_{t+1} \sim \pi(\cdot|s_{t+1}), \forall t \geqslant 0}\left[\sum_{t=0}^{\infty} \gamma^t r(s_t, a_t) | s_0 = s, a_0 = a\right],$$

and satisfies the Bellman equation (Bellman, 1966)

$$Q^\pi(s, a) = r(s, a) + \gamma \mathbb{E}_{s' \sim P(\cdot|s,a), a' \sim \pi(\cdot|s')} \left[ Q^\pi(s', a') \right].$$

**Policy Mirror Descent.** We focus on extracting policies from a learned state-action value function $Q^{\pi_{\text{old}}}(s, a) = \mathbb{E}_{\pi_{\text{old}}}[\sum_{\tau=0}^{\infty} \gamma^\tau r(s_t, a_t)|s_0 = s, a_0 = a]$ of the current policy $\pi_{\text{old}}$. We consider policy mirror descent with Kullback–Leibler (KL) divergence proximal term (Lan, 2023; Tomar et al., 2021; Vieillard et al., 2020; Abdolmaleki et al., 2018; Peters et al., 2010), which updates the policy with

$$\pi(a|s) \coloneqq \operatorname*{argmax}_{\pi:\mathcal{S}\to\Delta(\mathcal{A})} \mathbb{E}_{a\sim\pi} \left[ Q^{\pi_{\text{old}}}(s, a) \right] - \lambda D_{KL} \left( \pi || \pi_{\text{old}}; s \right) \tag{1}$$

The additional KL divergence objective constrains the updated policy to be approximately within the trust region. Policy mirror descent is closely related to practical proximity-based algorithms such as TRPO (Schulman, 2015) and PPO (Schulman et al., 2017), but with a different approach to enforce the proximity constraints. The closed-form solution of policy mirror descent in (1) satisfies

$$\pi(a|s) = \pi_{\text{old}}(a|s) \frac{\exp\left(Q^{\pi_{\text{old}}}(s, a)/\lambda\right)}{Z(s)}, \tag{2}$$

and $Z(s) = \int \pi_{\text{old}}(a|s) \exp\left(Q(s, a)/\lambda\right) da$ is the partition function or normalization constant.

**Flow Matching and Mean Flow** Flow-based generative models (Lipman et al., 2022; Liu et al., 2022; Albergo & Vanden-Eijnden, 2023) define time-dependent vector field $v : [0, 1] \times \mathbb{R}^d \to \mathbb{R}^d$ that constructs a probability density path $p : [0, 1] \times \mathbb{R}^d \to \mathbb{R}_{>0}$. The target distribution $p_1(x_1)$ can be generated by first sampling from the tractable distribution $p_0(x_0)$, and then solving the ordinary differential equation (ODE) $\frac{dx_t}{dt} = v_t(x_t)$. Flow matching (Lipman et al., 2022) provides an efficient and scalable way to learn the velocity field $v$ by minimizing the following objective,

$$L_{\text{CFM}}(\theta) \coloneqq \mathbb{E}_{x_0 \sim p_0, x_1 \sim p_1, t \sim \mathcal{U}[0,1]} \|(x_1 - x_0) - v_\theta(t, x_t)\|_2^2 \tag{3}$$

with the straight interpolation $x_t = tx_1 + (1 - t)x_0$.

MeanFlow (Geng et al., 2025) was recently proposed to avoid the iterative sampling process in flow matching and enable the one-step generative modeling. Instead of using instantaneous velocity, MeanFlow characterizes flow fields with average velocity $u(x_t, r, t) \triangleq \frac{\int_r^t (v(x_\tau, \tau)) d\tau}{t - r}$. The average velocity field is learned with the variational iteration loss

$$L_{\text{MF}}(\theta) \coloneqq \mathbb{E}_{x_t, r, t} \|u_\theta(x_t, r, t) - \text{sg}(u_{\text{tgt}})\|_2^2, \tag{4}$$

$$\text{where} \qquad u_{\text{tgt}} = v(x_t, t) - (t - r)(v(x_t, t)\partial_x u_\theta + \partial_t u_\theta), \tag{5}$$

and $\text{sg}(\cdot)$ denotes the stop-gradient operation to avoid higher-order gradient calculation. Once we learn the average velocity field $u$, sampling of $p_1$ is performed in one step with $x_1 = x_0 + u(x_0, 0, 1)$, where $x_0$ is sampled from the tractable prior distribution $p_0$.

## 3 ONE-STEP EFFICIENT INFERENCE FOR FLOW POLICY

In this section, we introduce online RL policy learning method for flow and MeanFlow policy, both of which enable 1-step fast sampling for policy inference. We first derive a tractable loss function for online RL with flow policy in Section 3.1, and then establish an upper bound on the one-step sampling error in Section 3.2. In Section 3.3, we propose online RL training for the MeanFlow policy, along with a convergence guarantee under mild assumptions on the MeanFlow operator.

### 3.1 POLICY MIRROR DESCENT WITH FLOW MODEL

We parametrize the policy as a flow model that transport the simple Gaussian distribution $a_0 \sim \mathcal{N}(\mu, \sigma^2)$ to the target distribution as the solution to policy mirror descent $a_1 \sim \pi_{\text{old}}(a_1|s) \exp\left(Q^{\pi_{\text{old}}}(s, a_1)/\lambda\right)/Z(s)$, where $Z(s) = \int \pi_{\text{old}}(a_1|s) \exp\left(Q^{\pi_{\text{old}}}(s, a_1)/\lambda\right) da_1$. The corresponding velocity field $v(a_t, t|s)$ can be learned by minimizing the flow matching loss in Equation (3):

$$\mathbb{E}_{a_0 \sim \mathcal{N}(\mu, \sigma^2), a_1 \sim \pi_{\text{old}}(a_1|s) \exp(Q^{\pi_{\text{old}}}(s, a_1)/\lambda)/Z(s), t \sim \mathcal{U}[0,1]} \|(a_1 - a_0) - v_\theta(a_t, t|s)\|^2. \tag{6}$$

One major difference between our case and flow matching in image generation or imitation learning is that we do not have access to direct samples from the target distribution of $a_1$. Consequently, standard flow matching cannot be applied directly to learn the target policy distribution. To solve this issue, we apply importance sampling to Equation (6) and sample from the base distribution $\pi_{\text{old}}$ to get a per-state loss function:

$$\tilde{L}_{\text{FPMD}}(\theta; s) \coloneqq \mathbb{E}_{a_1 \sim \pi_{\text{old}}(a_1|s), a_0 \sim \mathcal{N}, t \sim \mathcal{U}[0,1]} \left[ \frac{\exp(Q^{\pi_{\text{old}}}(s,a_1)/\lambda)}{Z(s)} \|(a_1 - a_0) - v_\theta(a_t, t|s)\|^2 \right] \quad (7)$$

Note that for each fixed $s$, $Z(s) = \int \pi_{\text{old}}(a|s) \exp(Q(s,a)/\lambda) \, da > 0$, and in Equation (7), $Z(s)$ is a constant independent of $\theta, a_1, a_0$ and t. Multiplying an objective by a positive constant does not change its minimizer, i.e.,

$$\operatorname*{argmin}_\theta \tilde{L}_{\text{FPMD}}(\theta; s) = \operatorname*{argmin}_\theta Z(s) \tilde{L}_{\text{FPMD}}(\theta; s), \forall s \in \mathcal{S}. \quad (8)$$

Take the expectation of $Z(s) \tilde{L}_{\text{FPMD}}(\theta; s)$ over $\mathcal{S}$, we obtain the practical flow policy learning loss,

$$L_{\text{FPMD}}(\theta) \coloneqq \mathbb{E}_{s, a_1 \sim \pi_{\text{old}}(a_1|s), a_0 \sim \mathcal{N}, t \sim \mathcal{U}[0,1]} \left[ \exp(Q^{\pi_{\text{old}}}(s, a_1)/\lambda) \|(a_1 - a_0) - v_\theta(a_t, t|s)\|^2 \right]. \quad (9)$$

Equation (9) gives a feasible loss function definition that only requires sampling state $s$ from replay buffer, action $a_0$ from a Gaussian distribution, and action $a_1$ from the old policy $\pi_{\text{old}}$. This loss formulation enables the use of replay buffer to estimate the loss function, which can efficiently learn the velocity field by minimizing the loss function using stochastic gradient descent.

### 3.2 ONE-STEP SAMPLING OF FLOW POLICY AND DISCRETIZATION ERROR BOUND

One-step sampling is attractive in practice because it significantly reduces the inference latency and computational cost in flow policy execution. In this subsection, we prove that the one-step sampling discretization error of the flow policy in Section 3.1 is bounded by the variance of the target policy distribution. Since the policy mirror descent target distribution typically converges to an almost-deterministic distribution (Puterman, 2014; Johnson et al., 2023)(Also see Appendix C.2 for empircial evidence), the policy variance becomes small, which provably guarantees a small discretization error. This property enables efficient one-step inference of flow policy without extra distillation (Park et al., 2025) or consistency loss (Ding & Jin, 2023).

**Proposition 1.** *[Proposition 3.3, (Hu et al., 2024)] Define $p_t^*$ as the marginal distribution of the exact ODE $da_t = v(a_t, t|s)dt$. Assume $a_t \sim p_t = p_t^*$, and let $p_{t+\epsilon_t}$ be the distribution of $a_{t+\epsilon_t}$ following $a_{t+\epsilon_t} = a_t + \epsilon_t v(a_t, t|s)$, where $\epsilon_t \in [0, 1-t]$ is a discretization step size. Then we have*

$$W_2\left(p_{t+\epsilon_t}^*, p_{t+\epsilon_t}\right)^2 \leqslant \epsilon_t^2 \mathbb{E}_{a_t \sim p_t}\left[\sigma^2(a_t, t|s)\right],$$

*where $\sigma^2(a_t, t|s) = var(a_1 - a_0|a_t, s)$, $p_{t+\epsilon_t}^*$ denotes the marginal distribution of the exact ODE at time $t + \epsilon_t$, and $W_2$ denotes the 2-Wasserstein distance.*

This proposition establishes the relationship between the sampling discretization error $W_2\left(p_{t+\epsilon_t}^*, p_{t+\epsilon_t}\right)^2$ and the conditional variance $\sigma^2(a_t, t|s)$. As a special case, when the target distribution $a_1$ has zero variance, the discretization error $W_2\left(p_{t+\epsilon_t}^*, p_{t+\epsilon_t}\right)^2 = 0$ for any $\epsilon_t \in [0, 1-t]$ (Hu et al., 2024). We then focus on the single-step sampling case and obtain the following result.

**Proposition 2.** *Define $p_t^*$ as the marginal distribution of the exact ODE $da_t = v(a_t, t|s)dt$. Let $p_1$ be the distribution of $\hat{a}_1$ such that $\hat{a}_1 = a_0 + v(a_0, 0|s)$ using one-step sampling, then*

$$W_2(p_1^*, p_1)^2 \leqslant var(a_1|s).$$

*Proof.* Take $t = 0$ and $\epsilon_t = 1$ in Proposition 1, we obtain that

$$W_2(p_1^*, p_1)^2 \leqslant \mathbb{E}_{a_0}\left[var(a_1 - a_0|a_0, s)\right] = \mathbb{E}_{a_0}\left[var(a_1|a_0, s)\right] = \mathbb{E}_{a_0}\left[var(a_1|s)\right] = var(a_1|s),$$

which concludes the proof of Proposition 2. $\qquad\square$

The proposition implies the discretization error in one-step sampling is bounded by the variance of the target distribution. In practical scenarios, where the learned policy tends to converge to one deterministic solution with small variance, the one-step discretization error is neglectable.

### 3.3 POLICY MIRROR DESCENT WITH MEANFLOW MODEL

In this subsection, we propose an alternative parametrization of the policy model using MeanFlow model (Geng et al., 2025). Before presenting the MeanFlow policy, we first introduce the fixed-point iteration view of MeanFlow, which fill the hole in the MeanFlow training (Geng et al., 2025). Specifically, although the MeanFlow Identity is the condition derived in Geng et al. (2025) for mean velocity, due to the *stop-gradient operator* induced in optimization for stability, it is not clear whether the optimization is still converging to target solution. The revealed fixed-point view does not only suggest the condition to guarantee the convergence of MeanFlow learning, but also justifies our MeanFlow policy mirror descent method.

Consider the velocity field $v(a_t, t|s)$ transporting $a_0 \sim \mathcal{N}\left(\mu, \sigma^2\right)$ to the policy mirror descent closed-form solution $a_1 \sim \pi_{\text{old}}\left(a_1|s\right) \exp\left(Q^{\pi_{\text{old}}}\left(s, a_1\right)/\lambda\right)/Z(s)$, we define the mean velocity field $u\left(a_t, r, t|s\right) \triangleq \frac{\int_r^t v(a_\tau, \tau|s)d\tau}{t-r}$. We define the MeanFlow operator applied to $u$ under the given $v$:

**MeanFlow operator:** $\quad (\mathcal{T}u)(a_t, r, t|s) = v(a_t, t|s) - (t-r)(v(a_t, t|s)\partial_a u + \partial_t u).$ $\quad$ (10)

Then, the original MeanFlow algorithm (Geng et al., 2025) can be viewed as variational implementation of the functional update $u_n = \mathcal{T}(u_{n-1})$ (Wen et al., 2020). Concretely, the MeanFlow operator is defined in a functional space with an unknown velocity field $v$, which is intractable to implement. To develop a practical algorithm, in Proposition 3 we propose a variational method that considers a reformulated problem whose optimal solution is equivalent to $\mathcal{T}(u_{n-1})$.

**Proposition 3.** *Given the previous iteration result $u_{n-1}$, define the residual loss*

$$L_{CMF}\left(\theta_n; s\right) = \mathbb{E}_{a_0, a_1, r, t}\left\|u_{\theta_n}(a_t, r, t|s) - \left((a_1 - a_0) - (t-r)\left((a_1 - a_0)\partial_a u_{n-1} + \partial_t u_{n-1}\right)\right)\right\|^2,$$
$$(11)$$

*where $a_t = ta_1 + (1-t)a_0$. The optimal solution $u^*_{\theta_n}(a_t, r, t|s) = \arg\min_{\theta_n} L_{CMF}\left(\theta_n; s\right)$ matches to the MeanFlow operator result $\mathcal{T}(u_{n-1})$.*

*Proof.* We characterize the optimality through the first-order condition, *i.e.*,

$\nabla_{\theta_n} L_{\text{CMF}}\left(\theta_n; s\right)$

$= \underset{a_0, a_1, r, t}{\mathbb{E}}\left[2\left(u_{\theta_n}(a_t, r, t|s) - \left((a_1 - a_0) - (t-r)\left((a_1 - a_0)\partial_a u_{n-1} + \partial_t u_{n-1}\right)\right)\right)\nabla_{\theta_n}u_{\theta_n}\left(a_t, r, t|s\right)\right]$

$= \underset{a_t, r, t}{\mathbb{E}}\underset{a_0, a_1|a_t}{\mathbb{E}}\left[2\left(u_{\theta_n}(a_t, r, t|s) - \left((a_1 - a_0) - (t-r)\left((a_1 - a_0)\partial_a u_{n-1} + \partial_t u_{n-1}\right)\right)\right)\nabla_{\theta_n}u_{\theta_n}\left(a_t, r, t|s\right)\right]$

$= \underset{a_t, r, t}{\mathbb{E}}\left[2\left(u_{\theta_n}(a_t, r, t|s) - \left(\mathbb{E}\left[X_1 - X_0|X_t = a_t\right] - (t-r)\left(\mathbb{E}\left[X_1 - X_0|X_t = a_t\right]\partial_a u_{n-1} + \partial_t u_{n-1}\right)\right)\right)\right.$

$\left. \nabla_{\theta_n}u_{\theta_n}\left(a_t, r, t|s\right)\right]$

$= \nabla_{\theta_n}\underbrace{\mathbb{E}_{a_t, r, t}\left[\left\|u_{\theta_n}\left(a_t, r, t|s\right) - \left(v(a_t, t|s) - (t-r)\left(v(a_t, t|s)\partial_a u_{n-1} + \partial_t u_{n-1}\right)\right)\right\|^2\right]}_{L_{\text{MF}}(\theta_n; s)}$ $\quad$ (12)

Therefore, we can show that the optimal solution $u^*_{\theta_n}$ satisfies:

$$u^{*(\text{CMF})}_{\theta_n}(a_t, r, t|s) = u^{*(\text{MF})}_{\theta_n}(a_t, r, t|s) = v(a_t, t|s) - (t-r)\left(v(a_t, t|s)\partial_a u_{n-1} + \partial_t u_{n-1}\right). \quad (13)$$

$\qquad\qquad\qquad\qquad\qquad\qquad\qquad\qquad\qquad\qquad\qquad\qquad\qquad\qquad\qquad\qquad\qquad\qquad\square$

Consequently, this fix-point iteration view of MeanFlow immediately induces the sufficient condition to guarantee the convergence to target distribution, following the fixed-point theorem (Banach, 1922), *i.e.*,

**Proposition 4.** *If the MeanFlow operator $\mathcal{T}$ satisfies the Contraction Condition, i.e., $\exists q \in [0, 1)$ such that $\|\mathcal{T}(u_1) - \mathcal{T}(u_2)\| \leqslant q\|u_1 - u_2\|$ for $\forall u_1, u_2 \in L^2$, then, with any initial point $u_0 \in L^2$, the fix-point iteration $u_n = \mathcal{T}(u_{n-1})$ for $n \geqslant 1$ converges to $u^*$ with $u^* = \mathcal{T}(u^*)$, which satisfies the MeanFlow Identity in (Geng et al., 2025).*

With the target distribution convergence justified, we exploit importance sampling to avoid direct sampling from target distribution $\pi(a|s) = \pi_{\text{old}}(a|s)\frac{\exp(Q^{\pi_{\text{old}}}(s, a)/\lambda)}{Z(s)}$, which leads to the following result.

**Theorem 5** (MeanFlow Policy Mirror Descent). *By sequentially minimizing the loss*

$$L_{MPMD}(\theta_n; s) := \mathbb{E}_{a_0, r, t, a_1 \sim \pi_{old}} \Big[ \frac{\exp(Q^{\pi_{old}}(s, a_1)/\lambda)}{Z(s)}$$

$$\|u_{\theta_n}(a_t, r, t|s) - ((a_1 - a_0) - (t - r)((a_1 - a_0)\partial_a u_{n-1} + \partial_t u_{n-1}))\|^2 \Big], \quad (14)$$

*for $n = 1, 2, \ldots$, the learned $u_{\theta_n}^*(a_t, r, t|s)$ converges to the mean velocity field $u(a_t, r, t|s) = \frac{\int_r^t v(a_\tau, \tau|s)d\tau}{t - r}$ if $\mathcal{T}$ satisfies the Contraction Condition.*

The full proof of Theorem 5 is provided in Appendix B.1. Since the learned mean velocity field $u_\theta(a_t, r, t|s)$ converges to the true mean velocity field $u(a_t, r, t|s)$, we recover the target policy distribution $a_1$ by first sampling $a_0 \sim \mathcal{N}(\mu, \sigma^2)$, and then setting $a_1 = a_0 + u_\theta(a_0, 0, 1|s)$.

We take the expectation w.r.t. $L_{\text{MPMD}}(\theta_n; s)$ over $\mathcal{S}$ and obtain the following practical policy learning loss:

$$L_{\text{MPMD}}(\theta_n) := \mathbb{E}_{s, r, t} \mathbb{E}_{a_1 \sim \pi_{old}(a_1|s), a_0 \sim \mathcal{N}} \Big[ \exp(Q^{\pi_{old}}(s, a_1)/\lambda)$$

$$\|u_{\theta_n}(a_t, r, t|s) - ((a_1 - a_0) - (t - r)((a_1 - a_0)\partial_a u_{n-1} + \partial_t u_{n-1}))\|^2 \Big]. \quad (15)$$

Although the flow policy in Section 3.1 can achieve one-step sampling during inference of a trained policy, it still requires multiple sampling steps to obtain high-quality samples from $\pi_{old}$ during training. MeanFlow policy reduces this computational cost by using one-step sampling throughout the training process.

## 4 FLOW POLICY MIRROR DESCENT ALGORITHM

In this section, we introduce Flow Policy Mirror Descent (FPMD), a practical off-policy RL algorithm achieving strong expressiveness, efficient training and efficient inference. We present two variants, FPMD-R and FPMD-M, using the flow and MeanFlow policy parametrization described in 3 respectively. An overview of our algorithm is provided in Algorithm 1.

---

**Algorithm 1** Flow Policy Mirror Descent (FPMD)

---

**Require:** initial policy $\pi_\theta$, Q-function $Q_\phi$, replay buffer $\mathcal{D} = \emptyset$, MDP $\mathcal{M}$, total epochs $T$
1: **for** epoch $e = 1, 2, \ldots, T$ **do**
2:     Interact with $\mathcal{M}$ using policy $\pi_\theta$ and update replay buffer $\mathcal{D}$
3:     Sample batch $\{(s, a, r, s')\} \sim \mathcal{D}$
4:     Sample $a'$ via flow sampling with $\pi_\theta$
5:     **Critic learning**: update $Q_\phi$ by minimizing double Q-learning loss in Equation (18)
6:     **Actor learning**: represent $\pi_\theta$ by flow policy (FPMD-R) or MeanFlow policy (FPMD-M):
   $\Big\{$ FPMD-R: Sample $a_0 \sim \mathcal{N}, t \sim \mathcal{U}[0, 1]$, sample $a_1$ via flow sampling with $\pi_\theta$,
7:      Update $\pi_\theta$ by minimizing Equation (9)
      FPMD-M: Sample $a_0 \sim \mathcal{N}$, sample $r, t$, sample $a_1$ via flow sampling with $\pi_\theta$,
       Update $\pi_\theta$ by minimizing Equation (15)
8: **end for**

---

**Actor-Critic Algorithm**  Our training follows the standard off-policy actor-critic paradigm:

- **Critic learning:** we employ clipped double Q-learning (Fujimoto et al., 2018) and use n-step return estimation (Barth-Maron et al., 2018) in visual control environments. See Appendix D.1 for more details.

- **Actor learning:** we parametrize the policy distribution with flow model in FPMD-R and Mean-Flow model in FPMD-M for flexible distribution modeling. The policy is updated to fit the policy mirror descent closed-form solution Equation (2) for policy improvement. At each iteration, we sample $a_1$ from $\pi_{old}(a_1|s)$ via flow sampling with the current policy network parameters, sample $a_0$ from Gaussian distribution, and then compute the practical loss in Equation (9) for FPMD-R or Equation (15) for FPMD-M to run gradient descent.

Table 1: Results on OpenAI Gym MuJoCo environments. Reported are the mean and standard deviation of the best evaluation returns, computed across 5 random seeds. Values highlighted in blue correspond to the method achieving the best result among all the algorithms, and values highlighted in green indicate the best result among all methods with NFE=1 sampling.

| | | HALFCHEETAH | REACHER | HUMANOID | PUSHER | INVERTEDPENDULUM |
|---|---|---|---|---|---|---|
| **Classic Model-Free RL** | PPO (NFE=1) | $4852 \pm 732$ | $-8.69 \pm 11.50$ | $952 \pm 259$ | $-25.52 \pm 2.60$ | $1000 \pm 0$ |
| | TD3 ((NFE=1) | $8149 \pm 688$ | $-3.10 \pm 0.07$ | $5816 \pm 358$ | $-25.07 \pm 1.01$ | $1000 \pm 0$ |
| | SAC (NFE=1) | $8981 \pm 370$ | $-65.35 \pm 56.42$ | $2858 \pm 2637$ | $-31.22 \pm 0.26$ | $1000 \pm 0$ |
| **Diffusion Policy RL** | DIPO (NFE=20) | $9063 \pm 654$ | $-3.29 \pm 0.03$ | $4880 \pm 1072$ | $-32.89 \pm 0.34$ | $1000 \pm 0$ |
| | DACER (NFE=20) | $11203 \pm 246$ | $-3.31 \pm 0.07$ | $2755 \pm 3599$ | $-30.82 \pm 0.13$ | $801 \pm 446$ |
| | QSM (NFE=20 $\times$ 32 [1]) | $10740 \pm 444$ | $-4.16 \pm 0.28$ | $5652 \pm 435$ | $-80.78 \pm 2.20$ | $1000 \pm 0$ |
| | QVPO (NFE=20 $\times$ 32) | $7321 \pm 1087$ | $-30.59 \pm 16.57$ | $421 \pm 75$ | $-129.06 \pm 0.96$ | $1000 \pm 0$ |
| | DPMD (NFE=20 $\times$ 32) | $11924 \pm 609$ | $-3.14 \pm 0.10$ | $6959 \pm 460$ | $-30.43 \pm 0.37$ | $1000 \pm 0$ |
| **Flow Policy RL** | **RF-1 (NFE=**1) | $10163 \pm 590$ | $-3.32 \pm 0.22$ | $6469 \pm 344$ | $-23.29 \pm 1.63$ | $1000 \pm 0$ |
| | **MF (NFE=**1) | $9917 \pm 698$ | $-3.34 \pm 0.15$ | $6030 \pm 664$ | $-23.08 \pm 0.58$ | $1000 \pm 0$ |
| | | ANT | HOPPER | SWIMMER | WALKER2D | INVERTED2PENDULUM |
| **Classic Model-Free RL** | PPO (NFE=1) | $3442 \pm 851$ | $3227 \pm 164$ | $84.5 \pm 12.4$ | $4114 \pm 806$ | $9358 \pm 1$ |
| | TD3 (NFE=1) | $3733 \pm 1336$ | $1934 \pm 1079$ | $71.9 \pm 15.3$ | $2476 \pm 1357$ | $9360 \pm 0$ |
| | SAC (NFE=1) | $2500 \pm 767$ | $3197 \pm 294$ | $63.5 \pm 10.2$ | $3233 \pm 871$ | $9359 \pm 1$ |
| **Diffusion Policy RL** | DIPO (NFE=20) | $965 \pm 9$ | $1191 \pm 770$ | $46.7 \pm 2.9$ | $1961 \pm 1509$ | $9352 \pm 3$ |
| | DACER (NFE=20) | $4301 \pm 524$ | $3212 \pm 86$ | $103.0 \pm 45.8$ | $3194 \pm 1822$ | $6289 \pm 3977$ |
| | QSM (NFE=20 $\times$ 32) | $938 \pm 164$ | $2804 \pm 466$ | $57.0 \pm 7.7$ | $2523 \pm 872$ | $2186 \pm 234$ |
| | QVPO (NFE=20 $\times$ 32) | $718 \pm 336$ | $2873 \pm 607$ | $53.4 \pm 5.0$ | $2337 \pm 1215$ | $7603 \pm 3910$ |
| | DPMD (NFE=20 $\times$ 32) | $5683 \pm 138$ | $3275 \pm 55$ | $79.3 \pm 52.5$ | $4365 \pm 266$ | $9360 \pm 0$ |
| **Flow Policy RL** | **RF-1 (NFE=**1) | $5378 \pm 78$ | $3255 \pm 86$ | $60.2 \pm 10.6$ | $3973 \pm 541$ | $9359 \pm 1$ |
| | **MF (NFE=**1) | $5461 \pm 147$ | $2865 \pm 603$ | $54.7 \pm 10.2$ | $4404 \pm 285$ | $9355 \pm 2$ |

**Number of Sampling Steps** Sampling step number significantly influences the inference speed and sample quality of flow models. For FPMD-R, we take 20 sampling steps during training to accurately model the potentially high-variance intermediate policy distributions. During evaluation, we switch to one-step sampling instead to test its efficient inference capability. For FPMD-M we use one sampling step in both training and evaluation due to the average velocity parametrization.

## 5 EXPERIMENTS

In this section, we present the empirical results of our proposed online RL algorithms FPMD-R and FPMD-M[2]. First, we demonstrate the superior performance and inference speed of FPMD with comparison to prior Gaussian and diffusion policy methods. For a comprehensive evaluation, we benchmark on both proprioceptive state observation Gym MuJoCo (Todorov et al., 2012) and visual observation DMControl (Tassa et al., 2018) environments. We then visualize the action sampling trajectory for an intuitive understanding.

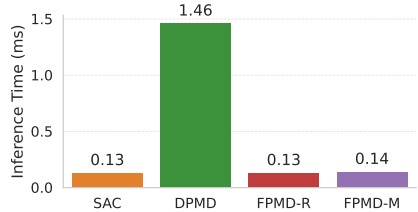

Figure 1: Policy inference time comparison between FPMD, Gaussian policy method SAC, and diffusion policy method DPMD.

### 5.1 GYM-MUJOCO TASKS

**Experiment Settings** We evaluate the performance on 10 Gym MuJoCo v4 environments. For all environments except Humanoid-v4, we train the policy for 200K iterations with 1M environment steps. For Humanoid-v4, we train for 1M iterations and 5M environment steps due to its more complex dynamics and higher-dimensional action space. We compare our method to two families of model-free online RL algorithms spanning both Gaussian and diffusion policy methods. See Appendix D.2 for more details.

**Comparative Evaluation** As shown in Table 1, FPMD achieves comparable performance with the best diffusion policy baseline while using $20\times$ fewer sampling steps during inference. Moreover, among methods using NFE (Number of Function Evaluations) =1 sampling, FPMD obtains the best overall performance.

---

[1]Here, 32 denotes the number of particles used for the best-of-N sampling mentioned in Section 4.

[2]Our implementation of FPMD can be found at `https://anonymous.4open.science/r/flow_policy_iclr-DD2B`.

Table 2: Evaluation on DMControl. The numbers show the mean and standard deviation of the best evaluation returns over 1M frames, computed across 5 random seeds.[3] Values highlighted in blue correspond to the method achieving the best result among all the model-free algorithms, and values highlighted in green indicate the best result among all model-free methods with NFE=1 inference sampling.

| | | CUP CATCH | WALKER WALK | CARTPOLE SWINGUP | FINGER SPIN |
|---|---|---|---|---|---|
| Gaussian Policy | SAC [4] (NFE=1) | $956.80 \pm 14.58$ | $865.29 \pm 67.17$ | $866.41 \pm 10.27$ | $927.09 \pm 62.33$ |
| | DDPG [5] (NFE=1) | $967.59 \pm 4.91$ | $215.29 \pm 413.94$ | $867.63 \pm 10.90$ | $752.10 \pm 178.04$ |
| Diffusion Policy | DPMD (NFE=20 × 32) | $979.70 \pm 1.91$ | $957.39 \pm 13.18$ | $843.23 \pm 17.00$ | $856.03 \pm 13.35$ |
| Flow Policy | **FPMD-R (NFE=1)** | $977.39 \pm 2.00$ | $957.05 \pm 5.39$ | $863.47 \pm 7.50$ | $862.29 \pm 42.98$ |
| | **FPMD-M (NFE=1)** | $974.96 \pm 7.05$ | $956.35 \pm 10.78$ | $849.88 \pm 7.56$ | $898.55 \pm 68.76$ |
| Model-based | DreamerV3 | $979.70 \pm 1.34$ | $967.28 \pm 3.76$ | $864.56 \pm 9.70$ | $622.25 \pm 164.38$ |
| | | CHEETAH RUN | DOG STAND | DOG TROT | DOG WALK |
| Gaussian Policy | SAC (NFE=1) | $507.70 \pm 40.32$ | $306.84 \pm 254.63$ | $92.60 \pm 22.31$ | $87.55 \pm 68.23$ |
| | DDPG (NFE=1) | $623.30 \pm 104.36$ | $321.26 \pm 200.80$ | $94.01 \pm 23.82$ | $109.33 \pm 52.31$ |
| Diffusion Policy | DPMD (NFE=20 × 32) | $631.74 \pm 32.43$ | $617.15 \pm 97.13$ | $113.93 \pm 56.68$ | $245.73 \pm 67.56$ |
| Flow Policy | **FPMD-R (NFE=1)** | $633.90 \pm 18.87$ | $599.92 \pm 168.44$ | $101.55 \pm 30.09$ | $221.08 \pm 137.77$ |
| | **FPMD-M (NFE=1)** | $619.97 \pm 51.30$ | $442.46 \pm 246.38$ | $94.10 \pm 35.99$ | $211.36 \pm 148.37$ |
| Model-based | DreamerV3 | $883.82 \pm 4.57$ | $542.12 \pm 295.74$ | $127.07 \pm 44.50$ | $139.54 \pm 12.51$ |

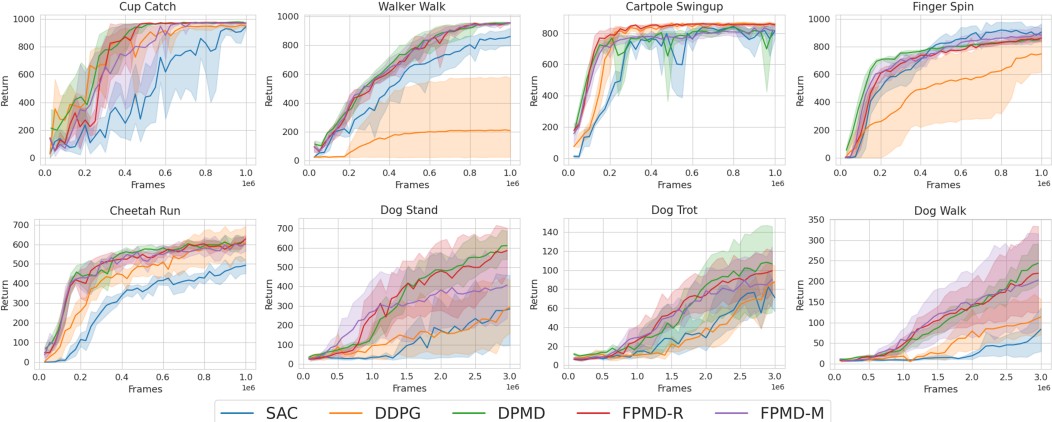

Figure 2: Performance curves on visual continuous control tasks. Shaded regions indicate the standard deviation over 5 random seeds.

To demonstrate the efficient policy inference ability of FPMD, we showcase the inference time in Figure 1. Inference speed was measured on a single RTX 6000 GPU and averaged over 10 rollouts on Ant-v4. All methods were implemented in JAX with JIT enabled and we performed one warm-start rollout that was not included in the measurements. As expected, both variants of FPMD achieve inference times more than $10\times$ faster than the diffusion policy method DPMD and match the speed of Gaussian policy.

## 5.2 VISUAL RL TASKS

We next evaluate FPMD on 8 visual-input continuous control tasks from the DeepMind Control Suite (Tassa et al., 2018). We compare FPMD against three policy learning algorithms: SAC (Haarnoja et al., 2018), DDPG (Lillicrap et al., 2015) and DPMD (Ma et al., 2025). The implementations of SAC and DDPG are based on DrQ (Kostrikov et al., 2020) and DrQ-v2 (Yarats et al.) respectively. As there are currently no online diffusion policy methods that achieve competitive results on visual DM-Control tasks, we implement a DPMD variant that aligns

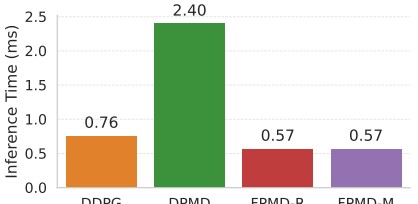

Figure 3: Policy inference time comparison between FPMD, Gaussian policy method DDPG, and diffusion policy method DPMD.

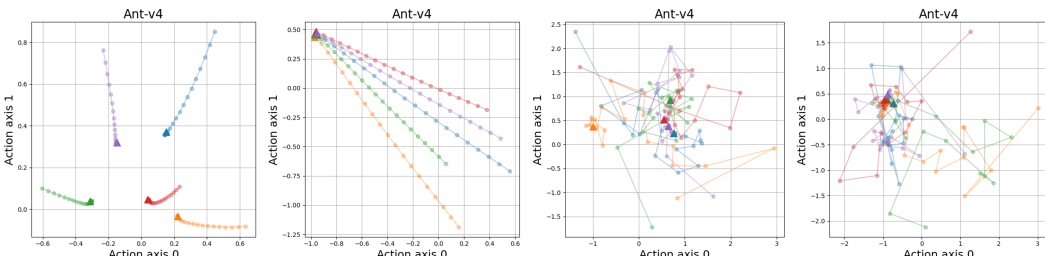

Figure 4: Sampling trajectories of `FPMD-R` and DPMD policy after 5K and 200K training iterations. From left to right: `FPMD-R` policy trained for 5K iterations, `FPMD-R` policy trained for 200K iterations, DPMD policy trained for 5K iterations, DPMD policy trained for 200K iterations.

with `FPMD` in all components except the policy learning.
As model-based algorithm use additional model learning losses and more training computation, it is not strictly fair to compare model-free and model-based algorithms. We only include the DreamerV3 (Hafner et al., 2023) results for reference.

Table 2 and Figure 2 show the benchmarking results on visual control tasks. `FPMD` achieves higher overall performance than all model-free gaussian policy baselines, with significant improvement in the most challenging dog domain. While matching the performance of the diffusion policy method DPMD, `FPMD` requires far fewer sampling steps and achieves over $4\times$ faster inference during evaluation, as shown in Figure 3.

Comparing our two flow policy variants `FPMD-R` and `FPMD-M`, we observe that the performance of `FPMD-M` slightly falls behind `FPMD-R` on all tasks except the relatively simple Finger Spin task. This gap could be due to the suboptimal action samples during training. Despite this, `FPMD-M` still has a clear advantage over the Gaussian policy baselines on most tasks. Since the MeanFlow policy parametrization can reduce the computational cost in sampling from $\pi_{\text{old}}$ during training, how to improve the performance of MeanFlow policy in visual control is a direction worth further investigation.

### 5.3 SAMPLING TRAJECTORY VISUALIZATION

We visualize the sampling trajectory to provide an intuitive demonstration of the small discretization error in single-step sampling of a well-trained `FPMD-R` policy. For comparison, we also visualize the sampling trajectory of the representative diffusion policy algorithm DPMD (Ma et al., 2025). We train the agents on Ant-v4 and plot the sampling trajectory of the first 2 action dimensions. Results are shown in Figure 4.

In the early stage of training, the policy mirror descent target distribution has a large variance, and the velocity varies significantly during the sampling process. Performing single-step sampling in this stage results in a large discretization error. By contrast, in the final stage of training, the target distribution exhibits small variance and the velocity is almost constant throughout the sampling process. Consequently, single-step sampling achieves small discretization error. However, for diffusion policy such as DPMD, directly performing single-step sampling would cause large error as demonstrated in Figure 4.

## 6 CONCLUSION

In this paper, we exploit the intrinsic connection between policy distribution variance and the discretization error of single-step sampling in straight interpolation flow matching. This insight leads to Flow Policy Mirror Descent (FPMD), an online RL algorithm that enables single-step sampling during policy inference while preserving expressive capability during training. We further present two algorithm variants based on flow and MeanFlow policy parametrizations respectively. Evaluation results on MuJoCo and DMControl benchmarks demonstrate performance comparable to diffusion policy methods while requiring two orders of magnitude less computational cost during inference. Future directions include extending FPMD to pretrained flow model finetuning and developing similar techniques for discrete decision-making domains.

ETHICS STATEMENT

This work focuses on reinforcement learning for flow policy, and our proposed method addresses the slow inference issue of flow policy via one-step sampling. All benchmarks used are publicly available. Our research does not involve human subjects and raises no specific ethical concerns requiring special attention.

REPRODUCIBILITY STATEMENT

We provide detailed descriptions of our method and training settings in Section 4 and Section 3. Additional implementation details, hyperparameters and evaluation protocol are available in Appendix D. The code for reproducing our results is provided in `https://anonymous.4open.science/r/flow_policy_iclr-DD2B`.

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

## A    RELATED WORK

**Reinforcement Learning for Diffusion Policy.**    Diffusion policies have been leveraged in many recent online RL studies due to their expressiveness and flexibility. To obtain tractable training objectives, existing methods explored reparameterized policy gradient (Wang et al., 2024; Ren et al., 2024; Celik et al., 2025), weighted self-improvement (Ma et al., 2025; Ding et al., 2024), and other variants of score matching (Psenka et al., 2023; Yang et al., 2023). However, these methods have not considered the inference-time difficulties of diffusion policies. A few recent studies on offline RL (Ding & Jin, 2023; Park et al., 2025) took a step toward one-step action generation. However, the solutions are complicated and involve multi-stage training.

**Reinforcement Learning for Flow Policy.**    Several concurrent works have explored reinforcement learning for flow policy via reparameterized policy gradient (Lv et al., 2025; Koirala & Fleming, 2025), reward-weighted regression (Pfrommer et al., 2025), and policy gradient with log-likelihood approximation (McAllister et al., 2025). Related approaches have also been proposed for offline RL (Zhang et al.), and finetuning pretrained flow policy and image generation models, including ORW-CFM-W2 (Fan et al., 2025), ReinFlow (Zhang et al., 2025), Flow-GRPO (Liu et al., 2025) and DanceGRPO (Xue et al., 2025). Compared to these concurrent methods, ours is the only method that achieves an effective balance between policy distribution expressiveness and action sampling efficiency, by introducing a practical training objective equivalent to the flow matching objective and enabling one-step action generation.

**Efficient Sampling of Flow Models.**    Although flow models are strong in modeling complex and multi-modal distributions, they typically require multiple sampling steps to generate high-quality samples (Lipman et al., 2022; Gat et al., 2024). To overcome this limitation, previous works have focused on producing high-quality samples in one- and few-step sampling settings. These methods mainly fall into the following two categories. The first category still learns a continuous velocity field but keeps the Euler truncation error small by either straightening the velocity field (Liu et al., 2022; 2023; Lee et al., 2024; Pooladian et al., 2023; Kornilov et al., 2024) or adjusting the sampling step size (Hu et al., 2024; 2023; Nguyen et al., 2023). The second category of methods distills the learned velocity field or directly learns the sampling trajectory, including CTM (Kim et al., 2023), shortcut model (Frans et al., 2024), and MeanFlow (Geng et al., 2025; Sheng et al., 2025). These efficient sampling strategies are orthogonal and compatible with our flow policies learned through online RL.

## B    DERIVATIONS

### B.1    PROOF OF THEOREM 5

**Theorem** (MeanFlow Policy Mirror Descent).    *By sequentially minimizing the loss*

$$L_{MPMD}(\theta_n; s) := \mathbb{E}_{a_0, r, t, a_1 \sim \pi_{old}} \Big[ \exp\left(Q^{\pi_{old}}(s, a_1)/Z(s)\right)$$

$$\left\| u_{\theta_n}(a_t, r, t|s) - ((a_1 - a_0) - (t - r)((a_1 - a_0)\partial_a u_{n-1} + \partial_t u_{n-1})) \right\|^2 \Big], \quad (16)$$

*for $n = 1, 2, \ldots$, the learned $u_{\theta_n}^*(a_t, r, t|s)$ converges to the mean velocity field $u(a_t, r, t|s) = \frac{\int_r^t v(a_\tau, \tau|s) d\tau}{t - r}$ if $\mathcal{T}$ satisfies the Contraction Condition.*

*Proof.* By substituting the target distribution $\pi_{old}(a_1|s)\exp\left(Q^{\pi_{old}}(s, a_1)/\lambda\right)/Z(s)$ into (11) and applying importance sampling, we obtain the per-state loss:

$$\tilde{L}_{MPMD}(\theta_n; s) := \mathbb{E}_{a_0, r, t, a_1 \sim \pi_{old}} \Big[ \exp\left(Q^{\pi_{old}}(s, a_1)/\lambda\right)/Z(s)$$

$$\left\| u_{\theta_n}(a_t, r, t|s) - ((a_1 - a_0) - (t - r)((a_1 - a_0)\partial_a u_{n-1} + \partial_t u_{n-1})) \right\|^2 \Big]. \quad (17)$$

Observe that for each fixed $s$, $Z(s) = \int \pi_{old}(a|s)\exp\left(Q(s, a)/\lambda\right) da > 0$ and is a constant independent of $\theta$, $a_1$, $a_0$, and $t$. Since multiplying an optimization objective by a positive constant does

not change its minimizer,

$$\operatorname*{argmin}_{\theta_n} \tilde{L}_{\text{MPMD}}(\theta_n; s) = \operatorname*{argmin}_{\theta_n} Z(s)\tilde{L}_{\text{MPMD}}(\theta_n; s) = \operatorname*{argmin}_{\theta_n} L_{\text{MPMD}}(\theta_n; s), \forall s \in \mathcal{S}.$$

Using Proposition 3,

$$u_{\theta_n}^{*(\text{MPMD})}(a_t, r, t|s) = u_{\theta_n}^{*(\text{CMF})}(a_t, r, t|s) = v(a_t, t|s) - (t - r)\left(v(a_t, t|s)\partial_a u_{n-1} + \partial_t u_{n-1}\right)$$

Then under Assumption 4, fixed point iteration theory implies

$$\lim_{n \to \infty} u_{\theta_n}^{*(\text{MPMD})}(a_t, r, t|s) = u(a_t, r, t|s) = \frac{\int_r^t v(a_\tau, \tau|s)d\tau}{t - r}$$

$\square$

# C ADDITIONAL RESULTS

## C.1 ABLATION STUDY

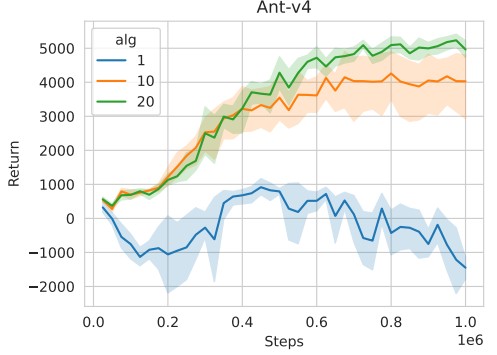
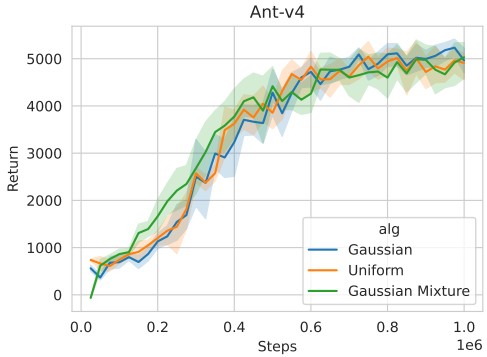

(a) Ablation study on training sampling steps.  (b) Ablation study on flow source distributions.

To assess how the number of sampling steps and source distribution affect policy performance, we conduct ablation experiments on the Gym MuJoCo Ant-v4 environment. In Figure 5a, we report results for different numbers of sampling steps when sampling from $\pi_{\text{old}}$ in `FPMD-R`. The results show that using too few sampling steps either fails to learn or results in suboptimal performance, likely due to the large discretization error in the early stage of training. We further study the effect of the source distribution by varying it across Gaussian distribution, bounded uniform distribution and a mixture of 2 Gaussians. As presented in Figure 5b, all the three variants perform similarly, so we select the commonly used Gaussian distribution for the main experiments.

## C.2 ADDITIONAL SAMPLING TRAJECTORY EXAMPLES

We provide additional `FPMD-R` action sampling trajectories in Figure 6. The flow policy is trained on Ant-v4 and we plot the sampling trajectories of the first 2 action dimensions throughout the training process. As shown in Figure 6, in the early stage of training the velocity learned by the policy network varies significantly in the sampling process, which leads to large discretization error for few step sampling. In contrast, once the policy is fully trained, the sampling velocity is nearly constant, enabling single step sampling with high accuracy.

# D EXPERIMENTAL DETAILS

## D.1 IMPLEMENTATION AND TRAINING DETAILS

**Network Architecture**  Both policy network and critic network in `FPMD` are MLPs with Mish (Misra, 2019) activations. We encode the flow time $t$ using the sinusoidal position embedding

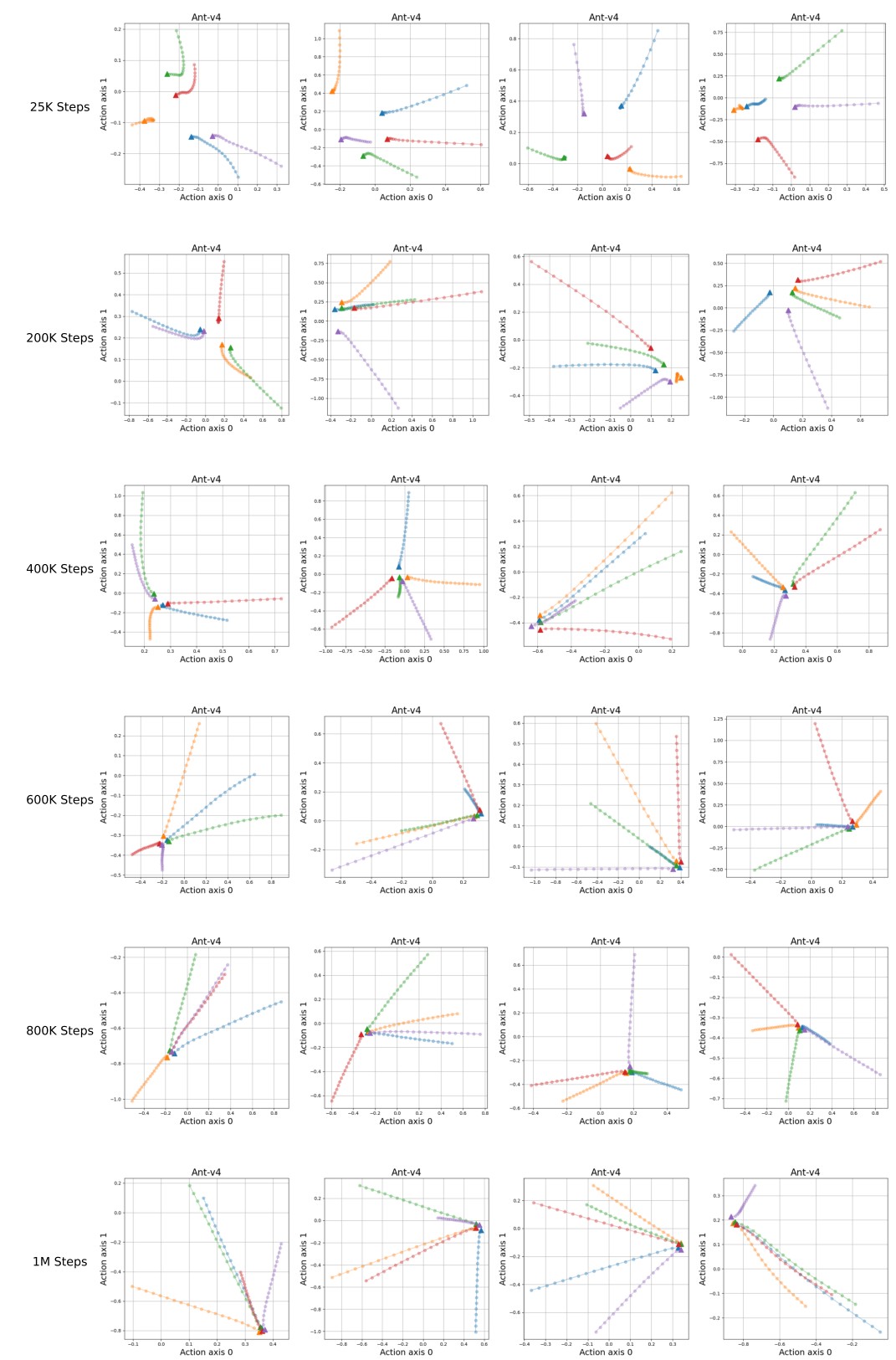

Figure 6: Sampling trajectories of FPMD-R on Ant-v4. At the beginning of training, the sampling trajectories are highly curved with non-uniform spacing between points of adjacent timesteps. As training proceeds, the trajectories become nearly straight with uniform spacing between points.

(Vaswani et al., 2017) and concatenate it to the policy network input. For visual control environments, we adopt the same convolutional encoder architecture as in Yarats et al. (2021), Kostrikov et al. (2020) and Yarats et al.. The visual encoder is updated using only gradient from the critic loss Equation (18).

**Selecting from Action Candidates**    Selecting from behavior candidates has been used in diffusion policy to concentrate the selected actions in high $Q$-value regions and improve sample efficiency (Chen et al., 2022; Ding et al., 2024; Ma et al., 2025). Instead of sampling only *one* action from flow policy, we sample $N$ actions for each state and select the action with the highest $Q$-value $a = \operatorname*{argmax}_{a_i} Q(s, a_i)$. We also add additional Gaussian noise with scheduled variance to the sampled actions for better exploration. During evaluation, we use the action sampled directly from the flow model with a single step to test the efficient inference ability.

**Critic Training**    For critic learning, we follow the common practice and employ clipped double Q-learning (Fujimoto et al., 2018) to reduce overestimation in the target value. We empirically find that in state-based environments n-step return estimation (Barth-Maron et al., 2018) leads to worse performance in flow policy, so we remain the 1-step setting. For visual control tasks, we use 3-step return estimation for faster reward propagation. Considering these empirical findings, the critic loss is

$$L_{\theta_k} = \mathbb{E}_{s,a,s',a'} \left[ \left( Q_{\theta_k}(s,a) - \left( r + \gamma \min_{k=1,2} Q_{\bar{\theta}_k}(s',a') \right) \right) \right] \quad \forall k \in \{1,2\}. \tag{18}$$

for state-based environments and Equation (19) for visual control environments.

**Visual Reinforcement Learning**    As is common in model-free visual RL algorithms (Yarats et al., 2021; Kostrikov et al., 2020), we encode raw pixel observations with a convolutional network and use the resulting latent feature as input to the policy and critic network. Following prior work (Kostrikov et al., 2020), we augment image observations with random image shifts and use the n-step critic loss

$$L_{\theta_k} = \mathbb{E}_{\{s_{t+i}, a_{t+i}\}_{i=0}^n \sim \mathcal{D}} \left[ \left( Q_{\theta_k}(s_t, a_t) - \left( \sum_{i=0}^{n-1} \gamma^i r_{t+i} + \gamma^n \min_{k=1,2} Q_{\bar{\theta}_k}(s_{t+n}, a_{t+n}) \right) \right) \right] \quad \forall k \in \{1,2\} \tag{19}$$

for faster reward propagation (Watkins et al., 1989). We ablate the n-step TD target and find it crucial for visual control tasks, especially in the most complex dog domain.

**Timestep Schedule**    Time schedule in flow and MeanFlow model learning has a large impact on the learned distribution as shown in prior work (Lee et al., 2024; Esser et al., 2024; Geng et al., 2025). For FPMD-R, we choose the commonly used uniform distribution $\mathcal{U}[0, 1]$. For the MeanFlow policy, we follow the default settings in (Geng et al., 2025) but change the $r \neq t$ probability from 25% to 100% to handle the online changing target distribution.

**Training and Evaluation Details**    We train for 1M environment frames by default, while for the most challenging tasks we increase the training steps. In particular, we use 5M environment frames on Humanoid-v4 and 3M frames on Dog Stand, Dog Trot and Dog Walk. Across all environments, we perform one training iteration every 5 collected environment steps. We report the average performance over 20 episodes for evaluation. For both FPMD-R and FPMD-M, we evaluate with single step sampling and no best-of-N sampling.

### D.2    BASELINES

**Gym Mujoco Environments**    We compare our method to two families of model-free online RL algorithms. The first family is classic RL algorithms with Gaussian policy parametrization, including PPO (Schulman et al., 2017), TD3 (Fujimoto et al., 2018) and SAC (Haarnoja et al., 2018). Those methods use 1-NFE (Number of Function Evaluations) sampling for action. The second family includes online diffusion policy algorithms DIPO (Yang et al., 2023), DACER (Wang et al., 2024), QSM (Psenka et al., 2023), QVPO (Ding et al., 2024) and DPMD (Ma et al., 2025). These methods require multiple sampling steps to generate high-quality actions in both training and inference. All baseline results reported for MuJoCo enviroments are taken from the DPMD paper (Ma et al., 2025).

**DMControl Environments** For visual environments, we compare `FPMD` against three policy learning algorithms: SAC (Haarnoja et al., 2018), DDPG (Lillicrap et al., 2015) and DPMD (Ma et al., 2025). The implementations of SAC and DDPG are based on DrQ (Kostrikov et al., 2020) and DrQ-v2 (Yarats et al.) respectively. As there are currently no online diffusion policy methods that achieve competitive results on visual DMControl tasks, we implement a DPMD variant that aligns with `FPMD` in all components except the policy learning.

### D.3 HYPERPARAMETERS

Table 3: Hyperparameters used for state-based Gym MuJoCo environments.

| Hyperparameter | Value |
|---|---|
| Critic learning rate | 3e-4 |
| Policy learning rate | 3e-4, linear annealing to 3e-5 |
| Value network hidden layers | 3 |
| Value network hidden neurons | 256 |
| Value network activation | Mish |
| Policy network hidden layers | 3 |
| Policy network hidden neurons | 256 |
| Policy network activation | Mish |
| Batch size | 256 |
| Replay buffer size | 1M |
| Action repeat | 1 |
| Frame stack | 1 |
| n-step returns | 1 |

Table 4: Hyperparameters used for visual observation DMControl environments.

| Hyperparameter | Value |
|---|---|
| Critic learning rate | 3e-4 |
| Policy learning rate | 3e-4, linear annealing to 3e-5 |
| Value network hidden layers | 3 |
| Value network hidden neurons | 256 |
| Value network activation | Mish |
| Policy network hidden layers | 3 |
| Policy network hidden neurons | 256 |
| Policy network activation | Mish |
| Encoder network convolutional layers | 4 |
| Encoder network kernel size | $3 \times 3$ |
| Encoder network activation | ReLU |
| Batch size | 256 |
| Replay buffer size | 1M |
| Action repeat | 2 |
| Frame stack | 3 |
| n-step returns | 3 |

## E   THE USE OF LLM

We used LLM for minor text polishing. The model did not contribute to research ideation, methodology, or results.

---

[3]We use 3M frames for tasks of the dog domain.

[4]Implementation based on DrQ-v1.

[5]Implementation based on DrQ-v2.

Table 5: `FPMD-R` Hyperparameters.

| Hyperparameter | Value |
|---|---|
| Sampling stepsize (training) | 0.05 |
| Sampling stepsize (evaluation) | 1.0 |
| t sampler | uniform(0, 1) |

Table 6: `FPMD-M` Hyperparameters.

| Hyperparameter | Value |
|---|---|
| Sampling stepsize (training) | 1.0 |
| Sampling stepsize (evaluation) | 1.0 |
| t, r sampler | uniform(0, 1) |

