# OpenReview forum: "One-Step Flow Policy Mirror Descent"
_ICLR.cc/2026/Conference — Submitted to ICLR 2026_

### Official Review · Reviewer_qRLo · 2025-10-30

**Soundness:** 3
**Presentation:** 3
**Contribution:** 3
**Rating:** 6
**Confidence:** 3

**Summary:**

Summary:
This work proposes Flow Policy Mirror Descent (FPMD), a one-step flow policy that is trained alongside the standard flow model. However, the trained one-step policy doesn't require any additional distillation or consistency training after training an ordinary diffusion/flow policy. Because the resulting policy generates actions with only a single denoising step, its inference time is reduced drastically. The effectiveness of the proposed method is demonstrated on various simulated online RL benchmarks, including image-based environments.

**Strengths:**

- This paper tackles an interesting and relevant approach in RL, more specifically w.r.t efficient policy representations with higher capacity than known Gaussian-based policies.

- The paper is well-motivated and has a nice story such that the reader can follow the steps to understand it.

**Weaknesses:**

- Please see the Questions below.

- Minor: I recommend running a grammar check. There are several grammatical issues, such as missing articles in the text.

**Questions:**

- The proposed method employs importance sampling in Eq. 7. However, importance sampling is known to have high variance, especially in higher-dimensional tasks/action spaces. Is there any analysis/observation regarding this fact? What is the highest action dimensionality considered in the experiments?

- Section 3.2 states that the target distribution converges to an almost deterministic distribution. Could the authors please provide an intuition for this? The optimal solution for the mirror descent policy is given as pi_old exp(Q(s,a)/\lambda), however, to my understanding, if \ lambda is rather large, this essentially means the new policy is very similar to the old policy. Doesn't this mean that it depends on how lambda is chosen rather than the converged solution?

- Could the paper provide an intuition behind Eq. 10? I think this could help improve the reader's understanding


- When comparing the results between classic model-free RL approaches to diffusion/flow-based approaches, it is interesting that for the gym environments, the diffusion/flow-based approaches seem to perform better (Table 1). However, looking at the vision-based environments, except for the dog environments, it looks like this gap is much smaller, or even closed for some environments, where the confidence intervals seem to overlap. Is there an intuition on this observation?

- Is lambda fixed, or optimized? If fixed, is there an intuitive way to choose its values? From my understanding, it depends on Q's value range. If optimized, how exactly is it optimized?


- The proposed framework does not consider the mostly used Maximum Entropy Reinforcement learning framework as used in the well-known SAC algorithm, but also used in diffusion-based policies such as proposed by O. Celik et al. 2025 (see citations in paper), where it seems that higher returns on the gym environments, namely the Ant and the more high-dimensional Humanoid environments for diffusion-based policies are reported. Is there a plan to compare against this objective variant as well?


- Connected to the question before, is there a possibility to extend the proposed framework to the Maximum entropy objective? How much does the assumption w.r.t. the deterministic, i.e., low variance target distribution, restrict the proposed method from applying it to the maximum entropy RL framework?

---

> ### Author Response · Authors · 2025-11-21
> **Response to Reviewer qRLo (Part 1)**
>
> We sincerely thank you for your valuable feedback on our submission. Below are our responses to the concerns you raised.
>
> **[Q1] Importance sampling is known to have high variance, especially in higher-dimensional tasks/action spaces. Is there any analysis/observation regarding this fact? What is the highest action dimensionality considered in the experiments?**
>
> In our experiments, the important sampling weighting ratio is $\frac{\pi _ {\mathrm{old}}(a _ 1|s)\exp\left(Q^{\pi _ {\mathrm{old}}}(s, a _ 1)/\lambda\right)/Z(s)}{\pi _ {\mathrm{old}}(a _ 1|s)}=\exp\left(Q^{\pi _ {\mathrm{old}}}(s, a _ 1)/\lambda\right)/Z(s)$. We normalize the $Q$-value in experiments and reduce the variance as the resulting importance weights have a smaller value range. The highest action dimension is 38 (Dog Stand, Dog Trot and Dog Walk). The training is stable in those high-dimensional tasks as shown in Figure 2.
>
> **[Q2] Section 3.2 states that the target distribution converges to an almost deterministic distribution. Could the authors please provide an intuition for this?**
>
> Our proposed FPMD follows the iterative training process in actor-critic online RL training. Each iteration will make the policy distribution more concentrated with the $\exp\left(Q^{\pi _ {\mathrm{old}}}(s, a)/\lambda\right)$ weighting function. The policy distribution in the $k$-th iteration is $\pi_k(a|s)=\pi _ 0(a|s)\exp\left(\sum_{i=0}^{k-1}Q^{\pi _ {\mathrm{i}}}(s, a)/\lambda\right)/Z _ k(s)$. We can observe that $\pi_k(a|s)\to \delta(\arg\max Q^\pi(s, a))$ when $k \to\infty$ and the converged distribution does not depend on $\lambda$.
>
> **[Q3] Could the paper provide an intuition behind Eq. 10?**
>
> Thank you for your suggestion. The MeanFlow training process in Geng et al. [1] can be viewed from the fixed point iteration view, and each iteration is performed by applying the MeanFlow operator in Equation (10) to $u _ {n-1}$. We define the MeanFlow operator to accurately describe this training process. We will update our paper to clarify this intuition.
>
> **[Q4] However, looking at the vision-based environments, except for the dog environments, it looks like this gap is much smaller, or even closed for some environments, where the confidence intervals seem to overlap. Is there an intuition on this observation?**
>
> Thank you for the careful observation. One possible reason is that, in some visual environments, the performance bottleneck lies primarily in encoder training, i.e., in learning a sufficiently rich latent representation, and the impact of using different policy distribution classes is relatively smaller. This can account for the reduced performance gap observed in some visual environments.
>
> **[Q5] Is lambda fixed, or optimized? If fixed, is there an intuitive way to choose its values?**
>
> Lambda is fixed throughout each training process. We choose lambda between 3.0 and 5.0 in all the experiments. The performance is robust to different choices of lambda with only small differences. As we normalize the $Q$-value in experiments, $\lambda$ mainly controls the exploration strength by shaping the sharpness of the target distribution. Empirically, we choose a relatively larger $\lambda$ (5.0) when the action dimension is large (e.g., tasks in the dog domain) for better exploration and choose a relatively small $\lambda$ (3.0) when the action dimension is small.
>
> [1] Geng Z, Deng M, Bai X, et al. Mean flows for one-step generative modeling[J]. arXiv preprint arXiv:2505.13447, 2025.

---

> > ### Author Response · Authors · 2025-11-21
> > **Response to Reviewer qRLo (Part 2)**
> >
> > **[Q6] The proposed framework does not consider the mostly used Maximum Entropy Reinforcement learning framework as used in the well-known SAC algorithm, but also used in diffusion-based policies such as proposed by O. Celik et al. 2025 (see citations in paper), where it seems that higher returns on the gym environments, namely the Ant and the more high-dimensional Humanoid environments for diffusion-based policies are reported.**
> >
> > We thank you for the suggestion to include a comparison with diffusion-based maximum entropy RL algorithms. Considering maximum entropy diffusion policy methods, both SDAC [2] and DIME [3] are formulated in the maximum entropy setting and learn diffusion policies to approximate the same target distribution $\frac{\exp{\left(Q(s, a)\right)}}{Z(s)}$. Since the environments used in our experiments are more closely aligned with those in the original SDAC paper, we therefore include in the table below a comparison between FPMD and SDAC on the state-based Gym MuJoCo v4 environments.
> >
> > |     | HalfCheetah| Reacher| Humanoid| Pusher| InvertedPendulum|
> > |---|---|---|---|---|---|
> > | SDAC   | $\textbf{12210} \pm \textbf{964}$| $-3.37 \pm 0.42$| $6437 \pm 177$| $-32.53 \pm 5.27$| $\textbf{1000} \pm \textbf{0}$|
> > | FPMD-R | $10163 \pm 590$| $\textbf{-3.32} \pm \textbf{0.22}$| $\textbf{6469} \pm \textbf{344}$| $-23.29 \pm 1.63$| $\textbf{1000} \pm \textbf{0}$|
> > | FPMD-M | $9917 \pm 698$| $-3.34 \pm 0.15$| $6030 \pm 664$| $\textbf{-23.08} \pm \textbf{0.58}$| $\textbf{1000} \pm \textbf{0}$|
> >
> > || Ant| Hopper| Swimmer| Walker2D| Inverted2Pendulum|
> > |---|---|---|---|---|---|
> > | SDAC   | $1391 \pm 202$| $2955 \pm 370$| $\textbf{119.1} \pm \textbf{41.9}$| $3995 \pm 498$| $\textbf{9360} \pm \textbf{0}$|
> > | FPMD-R | $5378 \pm 78$| $\textbf{3255} \pm \textbf{86}$| $60.2 \pm 10.6$| $3973 \pm 541$| $9359 \pm 1$|
> > | FPMD-M | $\textbf{5461} \pm \textbf{147}$| $2865 \pm 603$| $54.7 \pm 10.2$| $\textbf{4404} \pm \textbf{285}$| $9355 \pm 2$|
> >
> > We observe that the performance of FPMD is comparable to that of SDAC, while FPMD uses only a single step in policy inference, compared to 20 steps in SDAC. Please note that directly comparing the results in our paper with those in the DIME [3] paper is not entirely fair, as the experiment settings such as the MuJoCo environment version are not aligned.
> >
> > **[Q7] Connected to the question before, is there a possibility to extend the proposed framework to the Maximum entropy objective? How much does the assumption w.r.t. the deterministic, i.e., low variance target distribution, restrict the proposed method from applying it to the maximum entropy RL framework?**
> >
> > Thank you for your constructive question. As in maximum entropy RL, the optimal policy is not deterministic and usually has non-negligible variance, multiple sampling steps are required in FPMD-R to generate high-quality actions. For FPMD-M, low variance in the target distribution is not required for 1-step sampling, and we expect the mean flow variant to perform well in the maximum-entropy RL setting.
> >
> > [2] Ma H, Chen T, Wang K, et al. Efficient Online Reinforcement Learning for Diffusion Policy[J]. arXiv preprint arXiv:2502.00361, 2025.
> >
> > [3] Celik O, Li Z, Blessing D, et al. Dime: Diffusion-based maximum entropy reinforcement learning[J]. arXiv preprint arXiv:2502.02316, 2025.

---

### Official Review · Reviewer_GQsJ · 2025-11-01

**Soundness:** 3
**Presentation:** 2
**Contribution:** 2
**Rating:** 4
**Confidence:** 3

**Summary:**

This paper addresses a limitation of diffusion policies in online reinforcement learning -- their high inference cost. While diffusion models are highly expressive, they require a slow, iterative sampling process (many denoising steps) to generate an action, making them unsuitable for real-time applications.

The authors propose Flow Policy Mirror Descent (FPMD), an online actor-critic algorithm that uses a flow-based generative model as the policy. The core insight is a theoretical connection: in straight-interpolation flow models, the single-step sampling error is bounded by the variance of the target distribution. After the model converges, during exploitation, the optimal policy converges and typically has low variance. As this variance drops, the 1-step sampling error of the flow model also drops, allowing for highly efficient, single-step inference without any architectural changes, distillation, or consistency training.

To train this flow policy in an online RL setting, the authors derive a novel, practical loss function based on Policy Mirror Descent (PMD). This loss uses importance-weighted samples from the old policy to approximate the flow matching objective.

The paper presents two variants: FPMD-R (Rectified Flow) uses multi-step sampling during training but achieves 1-step sampling at inference. FPMD-M (MeanFlow) aims for 1-step sampling during both training and inference.

Experiments on MuJoCo and DeepMind Control Suite benchmarks show that FPMD achieves performance comparable to state-of-the-art diffusion policies while being 4x-10x faster at inference, matching the speed of simple Gaussian policies like SAC.

**Strengths:**

FPMD-R/M match or exceed diffusion policy baselines on most tasks.

The inference time speed-ups are clearly demonstrated in Figures 1 and 3.

The method is built on a solid foundation. Proposition 2 provides the theoretical bound for the 1-step error, and the derivation of the L_FPMD loss from the PMD objective appears sound.

**Weaknesses:**

The core importance-sampling loss (Eq. 9) is a relatively standard technique for applying generative models in reward-based learning, similar to reward-weighted objectives in image/video generation.

The core premise that the flow policy converges to an efficient 1-step model, relies on the optimal policy having low variance (being near-deterministic). This assumption, while true for the benchmarked tasks, may not hold for more complex or multi-task where the optimal policy itself is inherently stochastic or multi-modal. The paper's main benefit might not generalize to these settings.

 Following the points above, the paper could be viewed as a straightforward combination of a standard Actor-Critic algorithm with a Rectified Flow policy, primarily leveraging known properties of flow models rather than providing deep new insights into either RL or generative models.

**Questions:**

(Related to Weakness 2) How would FPMD perform in a task where the optimal policy is truly multi-modal (e.g., a scenario with two equally good but distinct solutions)? Would the policy "collapse" to a single mode to gain the 1-step sampling benefit, or would it maintain multi-modality and suffer a larger 1-step discretization error?

---

> ### Author Response · Authors · 2025-11-21
> **Response to Reviewer GQsJ**
>
> We sincerely thank you for your valuable feedback on our submission. Below are our responses to the concerns you raised.
>
> **[W1] The core importance-sampling loss (Eq. 9) is a relatively standard technique for applying generative models in reward-based learning.**
>
> The main point of Section 3.1 is to derive a practical loss function for online flow policy learning when **samples from the target distribution are not available** and **both the $Q$-function and the policy are iteratively updated**. To address these challenges in online RL, we first take the policy mirror descent approach, since it provides a closed-form policy iteration solution suitable for flow training. Then we apply the importance sampling technique to obtain a tractable loss that is equivalent to the flow matching loss. Although importance sampling is widely used in reward-based image generation model finetuning, Section 3.1 focuses on proposing a practical flow policy training loss in the online RL setting where both the $Q$-function and the policy are iteratively updated. We are also the first to demonstrate that this category of importance sampling based loss performs well in online reinforcement learning in both state-based and high-dimensional **visual control** tasks.
>
> **[W2] The core premise that the flow policy converges to an efficient 1-step model, relies on the optimal policy having low variance (being near-deterministic). This assumption, while true for the benchmarked tasks, may not hold for more complex or multi-task where the optimal policy itself is inherently stochastic or multi-modal. The paper's main benefit might not generalize to these settings.**
>
> For FPMD-R, we acknowledge that the multimodal policy may induce large discretization error in 1-step action sampling. However, for any MDP there always exists a deterministic optimal policy (Puterman [1]), and in practice the policy mirror descent target distribution typically converges to a low-variance distribution, which guarantees small discretization error for 1-step sampling of FPMD-R. On the other hand, the **FPMD-M** variant uses the MeanFlow parametrization and does not rely on this low variance property for 1-step sampling. As a result, it can enable high quality **1-step sampling regardless of the policy distribution's variance**, even when the learned policy is highly stochastic or multi-modal.
>
> **[W3] The paper could be viewed as a straightforward combination of a standard Actor-Critic algorithm with a Rectified Flow policy, primarily leveraging known properties of flow models rather than providing deep new insights into either RL or generative models.**
>
> This paper bridges the low variance property of policy mirror descent target distribution in online RL and the sampling discretization error property in straight-interpolation flow-based generative models. We then demonstrate how this connection can be used to enable 1-step efficient inference in flow policy without introducing additional training cost through both theoretical analysis and empirical evidence. This points out a free lunch of flow policy over diffusion policy in online RL, and has substantial potential impact in designing RL algorithms that balance policy expressiveness, efficient training and efficient inference.
>
> **[Q1] How would FPMD perform in a task where the optimal policy is truly multi-modal?**
>
> FPMD is compatible with tasks with multi-modal optimal policies. Concretely, our two variants behave as follows.
>
> **FPMD-R**. Our Proposition 2 provides a theoretical guarantee that FPMD-R has small 1-step sampling discretization error when the policy converges to a small-variance distribution, which is typically observed for policy mirror descent in common RL benchmarks. Moreover, for any MDP there always exists a deterministic optimal policy (Puterman [1]), so even when the set of optimal actions is multi-modal, there exists at least one deterministic optimal policy for which FPMD-R has small 1-step discretization error.
>
> **FPMD-M**. At the same time, our FPMD-M variant is designed not to rely on this low-variance property. It uses the MeanFlow policy parametrization, which enables high-quality **1-step action sampling regardless of the policy distribution's variance**, and thus naturally covers settings where the learned policy is highly stochastic or multi-modal.
>
> [1] Puterman M L. Markov decision processes: discrete stochastic dynamic programming[M]. John Wiley & Sons, 2014.

---

### Official Review · Reviewer_onHY · 2025-11-01

**Soundness:** 2
**Presentation:** 3
**Contribution:** 2
**Rating:** 4
**Confidence:** 4

**Summary:**

This paper introduces two expressive policies based on flow-based models that permit one-step sampling at evaluation time. The first model builds on rectified flow, which still requires multi-step sampling during training. For this model, the authors theoretically show that as the online policy’s stochasticity shrinks, the one-step sampling would have small error. Thus, after sufficient training, using the vector field at the first time step for all time steps will induce small errors, which is empirically demonstrated in a MuJoCo environment. On the other hand, the second model building on MeanFlow parameterizes the average velocity (for arbitrary window) directly, which permits one-step sampling in both training and evaluation time. Empirically, both models are shown to achieve competitive performance in state-based Gym MuJoCo and visual DeepMind Control Suite tasks, compared to diffusion-based policies.

**Strengths:**

The papers have several strengths:
1. The study problem is an interesting topic in reinforcement learning with continuous actions. With the recent development of expressive generative models, using them for policy representation is an interesting direction. The paper addresses an important limitation of current generative approaches – high inference time.
2. The paper reveals a novel insight that speeding up flow-based models with fewer sampling steps could be reasonable as the distribution becomes less stochastic, which aligns with the policy becoming more deterministic during training in RL. Such a property is not exploited in generative modeling as stochasticity is desirable and maintained there.
3. The paper is well-written. The mathematical notation is clear, and the text is easy to follow.

**Weaknesses:**

Despite the above strengths, there are also a few weaknesses of the paper:
1. Lack of controlled experiments to classic model-free RL approaches. The proposed methods perform very similarly to DPMD with their “proxy confidence intervals” overlapping with each other most of the time (note that a proper confidence interval should be used). It is reasonable to hypothesize that the actor update makes a relatively small influence on the training, which may not have been ruled out without further experiments (or clarifications on the current results).
2. Insufficient discussion of the limitations of the proposed approach.
  - By considering a partition function weighted loss for convenience, the matching objective puts more emphasis on states with higher action values, which is not necessarily ideal. The paper lack a discussion of potential consequences of doing so.
  - It does not discuss the training time of the proposed approaches. While evaluation (inference/test) time speed could be a crucial consideration, training time is also an important factor. The paper should disclose the training time for each model and compare it with other approaches.
3. While the fixed point view seems to be useful, the paper didn’t show the contraction property of the MeanFlow operator, which is a crucial fundamental property that guarantees a fixed point.
4. Some presentations of the empirical results could be misleading. 1) The highlight in the table does not take into account the randomness of experiments. Different methods often perform very similarly but only one of them is highlighted. 2) The claim in Lines 454 is based on results with high randomness and might not hold in general (e.g., for a different set of seeds).

**Questions:**

Could the authors clarify the below questions:
1. What do the shaded areas show in Figure 2?
2. Are there learning curves for the Gym MuJoCo experiments?
3. What’re the hyperparameters and critic updates for classic model-free RL methods in different settings? Are they consistent with the proposed methods?
4. Could the authors test classic model-free RL methods (SAC or DDPG) with the same configurations as the proposed methods whenever applicable? For example, the critic update, network configurations, best-of-n sampling, etc. should be used for them.

Other minor suggestions that do not impact the recommendation:
1. Line 045: distillation -> distilling
2. Line 139: mixed use of $a$ and $x$.
3. Line 142: the random variables should be written out explicitly in Eq. 4
4. Line 195: “…(is) the distribution…”

---

> ### Author Response · Authors · 2025-11-21
> **Response to Reviewer onHY**
>
> We sincerely thank you for your valuable feedback on our submission. Below are our responses to the concerns you raised.
>
> **[Q1] Lack of controlled experiments to classic model-free RL approaches.**
>
> For classic model-free RL algorithms, we include PPO, TD3, and SAC in state-based environments, and SAC and DDPG in visual environments. The proposed FPMD shows a clear advantage over these baselines in overall performance, as shown in Table 1 and Table 2.
> Note that DPMD [1] is a recently proposed online diffusion policy baseline instead of a classic model-free RL approach. The target policy distribution used in DPMD is the same as the policy mirror descent target distribution $\pi _ {\mathrm{old}}(a|s)\frac{\exp\left(Q^{\pi _ {\mathrm{old}}}(s, a)/\lambda\right)}{Z(s)}$ used in our method. It is both reasonable and empirically justified in the experiment results that our FPMD method performs similarly to the DPMD method, as the fitted policy distribution is the same. The main contribution in FPMD is that FPMD enables 1-step efficient inference (compared to the 20 sampling steps used in DPMD) while not introducing additional training cost.
>
> **[Q2] By considering a partition function weighted loss for convenience, the matching objective puts more emphasis on states with higher action values, which is not necessarily ideal.**
>
> We agree that this loss imbalance across states with different action values is a reasonable concern. One can use a separate network to predict the partition function or the value function, which would mitigate this issue. However, as we showed in Section 3.1, multiplying by the partition function does not change the optimal learned action distribution, and empirically we find that our proposed loss functions work well. Therefore, we do not include another network for estimating the partition or value function, in order to avoid additional computational and parameter cost.
>
> **[Q3] It does not discuss the training time of the proposed approaches.**
>
> Thank you for the suggestion. The training time comparison between our proposed FPMD and diffusion policy method DPMD, DACER is presented in the table below. We conduct the experiments by training for 200K iterations with 1M environment steps on the MuJoCo Ant-v4 environment. We will also include this result in the updated paper.
>
> ||DPMD|DACER|FPMD-R|FPMD-M|
> |---|---|---|---|---|
> |Training time (min)|21.52 |52.60|21.57|12.53|
>
> **[Q4] Other questions**
>
> >What do the shaded areas show in Figure 2?
>
> The shaded area shows the standard deviation over the 5 random seeds. We will clarify this in the updated paper.
>
> >Are there learning curves for the Gym MuJoCo experiments?
>
> We will include the training curves for the Gym MuJoCo environments in the paper.
>
> >What're the hyperparameters and critic updates for classic model-free RL methods in different settings? Are they consistent with the proposed methods?
>
> We keep all important hyperparameters consistent in both state-based and visual experiments. Please refer to Appendix D.3 for more details.
>
> >Could the authors test classic model-free RL methods (SAC or DDPG) with the same configurations as the proposed methods whenever applicable? For example, the critic update, network configurations, best-of-n sampling, etc. should be used for them.
>
> In our experiments, we already keep the main algorithm configurations of the classic model-free RL baselines aligned with those of FPMD whenever possible. This includes using the same critic update rules and network architectures, except in cases where such alignment would conflict with the original algorithm design (e.g., the entropy term in SAC critic learning).
> Regarding best-of-n sampling, since SAC requires log-probability calculation of the policy distribution during training, best-of-n sampling is not directly applicable. We therefore evaluate DDPG with best-of-n sampling during training and compare it with our method in the visual Cup Catch environment. As shown in the table below, best-of-n sampling leads to only a small change in the performance of DDPG, while our method still shows a clear advantage.
>
> ||DDPG|DDPG + best-of-n sampling| FPMD-R| FPMD-M|
> |---|---|---|---|---|
> |Cup Catch|$967.59 \pm 4.91$|$970.27 \pm 5.31$|$\textbf{977.39}\pm\textbf{2.00}$|$974.96\pm7.05$|
>
> > Minor issues
>
> Thank you for your feedback. We have fixed the mentioned minor issues in the paper.
>
> [1] Ma H, Chen T, Wang K, et al. Efficient Online Reinforcement Learning for Diffusion Policy[J]. arXiv preprint arXiv:2502.00361, 2025.

---

> > ### Comment · Reviewer_onHY · 2025-11-26
> >
> > Thank you for the rebuttal. I’ve read the responses and checked the relevant details the authors referred to. My concern regarding the training time is addressed. However, I have some follow-up questions on other concerns:
> >
> > > We keep all important hyperparameters consistent in both state-based and visual experiments. Please refer to Appendix D.3 for more details.
> >
> > I am aware of the hyperparameters listed in Appendix D.3. However, it does not explicitly mention which algorithms use these hyperparameters. For baseline results in MuJoCo, I notice that the DPMD repo uses GeLU in the critic network for SAC, while Mish is used for diffusion-based algorithms.
> > - Could the authors confirm the exact hyperparameters/configurations used for *model-free RL baselines*?
> > - To rigorously test the impact of the actor update, could you fully align the critic network architecture (e.g., activation functions), update frequencies, and other configurations across methods?
> >
> > https://github.com/mahaitongdae/diffusion_policy_online_rl/tree/main
> >
> > > As shown in the table below, best-of-n sampling leads to only a small change in the performance of DDPG, while our method still shows a clear advantage.
> >
> > Given the overlapping standard deviations, the performance difference does not appear to be statistically significant. Could the authors also perform experiments in walker-walk? Results on this environment might be more informative in distinguishing the methods. Again, it would be crucial to also compare with the version that aligns the critic network and update configurations here.
> >
> > > However, as we showed in Section 3.1, multiplying by the partition function does not change the optimal learned action distribution, and empirically we find that our proposed loss functions work well.
> >
> > While it does not change the optimal learned action distribution for the per-state objective, it will impact the optimal solution for the expected objective (Eq. 9) when $\pi$ is parameterized by a neural network. In this case, since the optimal action distribution might not be realizable for all states due to limited representation power (function approximation), the weighting of states will influence how the policy trades off errors across different states. I believe this discussion is necessary so readers are aware of the implications of the unweighted objective.
> >
> > > We will include the training curves for the Gym MuJoCo environments in the paper.
> >
> > I appreciate the authors’ effort and look forward to seeing it in the updated version.

---

> > > ### Author Response · Authors · 2025-12-04
> > >
> > > Thank you for taking the time to provide additional feedback and share your concerns with us. Below are our responses to the points you raised.
> > >
> > > Regarding the specific algorithm configurations, the hyperparameters listed in Table 3 and Table 4 are used for both FPMD and the diffusion policy baselines. For Gaussian policy baselines, we also keep all important hyperparameters such as network update frequencies aligned with the flow/diffusion policy methods whenever applicable, and will include these details in the updated paper.
> > >
> > > For best-of-n sampling, we conducted an additional experiment on its effect in DDPG on the Walker Walk environment and report the results in the table below.
> > >
> > > ||DDPG + best-of-n sampling| FPMD-R| FPMD-M|
> > > |---|---|---|---|
> > > |Walker Walk|$931.89 \pm 22.62$|$\textbf{957.05} \pm \textbf{5.39}$|$956.35\pm10.78$|
> > >
> > > From the results, we observe that after using best-of-n sampling in DDPG training, our flow policy method still achieves higher performance than the DDPG baseline.
> > >
> > > We thank the reviewer again for the constructive feedback and will update our paper accordingly.

---

### Official Review · Reviewer_fH6c · 2025-11-01

**Soundness:** 2
**Presentation:** 3
**Contribution:** 2
**Rating:** 2
**Confidence:** 4

**Summary:**

This paper presents a novel method to learn and evaluate flow-based policies that recover performance comparable to diffusion policy methods at a fraction of the computational cost. They adapt existing loss functions to the setting where the goal is to match a target mirror descent policy (the typical closed-form policy used for this setting) and take advantage of the fact that this target policy becomes more deterministic, reducing discretization error over time.

**Strengths:**

The paper is clearly written and easy to follow.

It is useful to investigate the new MeanFlow approach for learning policies in RL.

**Weaknesses:**

The paper does not sufficiently place this work relative to other papers in RL using flow policies in RL. The paper does cite several works and say:
“Compared to these methods, ours is the only method that achieves an effective balance between policy distribution expressiveness and action sampling efficiency, by introducing a practical training objective equivalent to the flow matching objective and enabling one-step action generation.”
However, this is an insufficient description of what is arguably the most important related work. The primary focus in the paper is on contrasting to diffusion policies. This is understandable, as the goal is to reduce the computational cost compared to this more popular strategy. However, it is key to better explain why existing alternatives do not already solve this problem. Further, at least one previous method from this literature should be included in the experiments, rather than only including diffusion approaches.

Throughout the paper there is also a tendency to restate previous results, without making it clear that is it not strictly new. Let me give a few examples.
1. The importance sampling approach used to sample from the mirror descent target policy is the standard approach (when using a mode-covering KL); though this sampling is never stated to be strictly new in Section 3.1, it is either implied or the reader is left unsure if this is a new strategy introduced here.

2. Proposition 1 is restated from Hu, used to then prove Proposition 2 that shows that discretization error is low if the variance is low. In their paper, Hu does talk about the ramifications of the variance being zero, so Prop 1 has been used before for somewhat similar reasoning. It would be better here to avoid restated Proposition 1, and potentially just writing you own new proposition, with some context for what Hu showed

3. Proposition 3 seems to focus on proving that minimizing LCFM is equivalent to minimizing LFM. But I believe this was already shown in (Yaron et al., 2022). Is that the case?

The final theory, Proposition 4 and Theorem 5, seems to state mostly basic facts about convergence of operators given contraction. However, the key criteria that is useful to prove here is that the contraction condition is satisfied for this operator. The current proof seems to largely restate what was in the main text, focused on how one samples and drops constants. Do you believe this is a contraction?

There is a key issue in the experiments that results are reported over 5 seeds. This is insufficient to make strong claims about performance. To make statistically significant claims, you could either run more seeds (justifying why that number is enough) or potentially aggregate over the environments and make claims at this aggregate level. For individual environments, you could look at individual runs to see behavior, rather than making strong claims about one method being better than another. One such strong claim is in Figure 2: “FPMD outperforms all baselines with NFE=1 sampling”. This cannot really be said, given the results there.

The paper also makes a claim that Mean Flow should be better when training under 1-step sampling compared to standard flow matching. This should be better demonstrated in your experiments. I did notice that in the appendix, you ran FPMD-R with fewer than 20 sampling steps during training. This is an important result and should be highlighted in the main body, and should also potentially include more than just 1, 10 and 20 sampling steps. It would also be useful to report training costs to show this tradeoff and better motivate FPMD-M, since otherwise FPMD-R and FPMD-M are relatively similar. Even better would be to also include another algorithm that uses flow matching, to identify if your specific objectives also provide improvements.

(Putting Minor Points here, so there is no separate box. These are not major issues)
- Typo on line 114: the citations need to be in brackets
- Same on lines 120, 121
- On line 418: “diffusion policy method SDAC” – this method is not mentioned anywhere else in the paper
- The acronym “NFE” is used a lot in the main text, but is only explained in the appendix (Number of Function Evaluations)
- “Important sammpling” on line 268
- Hanging “, which leads to” at the end of page 5
- The “rectified” flow is never defined, although the paper where it was originally introduced is cited.
- The claim “Although flow policy in Section 3.1 can achieve one-step sampling during inference of a trained policy, it still requires multiple sampling steps when sampling from πold during training. MeanFlow policy reduces this computational cost by using one-step sampling throughout the training process.” This sentence makes it sound like this is a requirement in the algorithm, when in reality this is due to empirical evidence suggesting that too few sampling steps in training result in suboptimal performance.
- The Peters et al citation is good for the KL loss, in Eq (1), but other works are more pertinent than the other two. For example, consider the MPO paper or a very nice overview paper called “Leverage the Average: an Analysis of KL Regularization in Reinforcement Learning”, Vieillard et al., NeurIPS 2020.

**Questions:**

1. Can you discuss in more detail how your approach differs from previous work using flow policies?

2. Can you provide some aggregation across environments, to make stronger claims, or justify why the number of seeds is sufficient?

3. Can you clarify the above questions on the theory? Is Proposition 3 new, or largely restating what is given in Yaron et al, and can you show the operator is a contraction?

4. Can you clarify the training stability + cost differences between FPMD-R and FPMD-M?
You present both learning curves and final performance tables for the visual tasks, but not the state-based tasks. Any reason for this?

5. “The numbers show the best mean returns and standard deviations over 1M frames and 5 random seeds.” – For the descriptions in tables 1 and 2, what does it mean to say the “best standard deviation”?

---

> ### Author Response · Authors · 2025-11-21
> **Response to Reviewer fH6c (Part 1)**
>
> We sincerely thank you for your valuable feedback on our submission. Below are our responses to the concerns you raised.
>
> **[Q1]. The paper does not sufficiently place this work relative to other papers in RL using flow policies in RL.**
>
> All the other online reinforcement learning for flow policy papers were published within the last four months before the ICLR full paper deadline and, as we pointed out in Line 92, should be considered as concurrent works. According to the ICLR policy,
> there is no requirement to compare our method against these concurrent baselines, and
> the paper's contribution should not be diminished by concurrent work.
>
> Despite this, considering the related concurrent works on online flow RL, Lv et al. [1] relies on maximizing the objective $\mathbb{E} _ {a\sim\pi _ \theta}\left[Q(s, a)\right]$ for policy improvement. It requires gradient backpropagation through multiple flow sampling steps to avoid performance degradation in general environments, which leads to large computation and memory cost. Koirala & Fleming [2] introduces an additional self-consistency loss for 1-step action sampling, and its policy loss is composed of three components, requiring loss weight tuning in practice and induces additional computational cost. Both Pfrommer et al. [3] and McAllister et al. [4] use multiple sampling steps (4 and 10 steps, respectively) in action generation, which results in slow inference.
>
> In contrast, our proposed FPMD uses a policy learning loss that is equivalent to the flow matching loss of the mirror descent target distribution, enabling expressive policy distribution modeling and efficient optimization during training.
> We also show that FPMD supports 1-step action sampling at inference time, as justified by both theoretical analysis and empirical results. Consequently, FPMD is the first online flow policy algorithm that simultaneously achieves strong policy expressiveness, efficient training, and efficient 1-step inference, without relying on distillation or additional consistency losses during training. We are also the first to demonstrate the effectiveness of online flow RL methods on more challenging **visual RL** tasks (Table 2 and Figure 2), rather than focusing solely on state-based tasks.
>
> **[Q2]. The importance sampling approach used to sample from the mirror descent target policy is the standard approach (when using a mode-covering KL); though this sampling is never stated to be strictly new in Section 3.1, it is either implied or the reader is left unsure if this is a new strategy introduced here.**
>
> The main point of Section 3.1 is to derive a practical loss function for online flow policy learning **when samples from the target distribution are not available**. To this end, we first take the policy mirror descent approach, since it provides a closed-form policy iteration solution suitable for flow training. Then we apply the standard importance sampling technique to obtain a tractable loss that is equivalent to the flow matching loss. Although importance sampling is a standard technique widely used in machine learning and we never describe it as a new technique in our paper, our main contribution in Section 3.1 focuses on proposing a practical online flow policy training loss for the setting where samples from the target distribution are not available, with importance sampling used only as one step in this derivation.
>
> **[Q3]. Proposition 3 seems to focus on proving that minimizing LCFM is equivalent to minimizing LFM. But I believe this was already shown in (Yaron et al., 2022). Is that the case?**
>
> Not exactly. Proposition 3 is not given in Yaron et al. [5] or Geng et al. [6]. While the intuition behind Proposition 3 is similar to the argument used in Yaron et al. [5] to show the equivalence between minimizing LCFM and LFM, our result extends this idea to the MeanFlow loss. In contrast, Yaron et al. [5] only established the equivalence in the original flow matching setting.
>
> [1] Lv L, Li Y, Luo Y, et al. Flow-Based Policy for Online Reinforcement Learning[J]. arXiv preprint arXiv:2506.12811, 2025.
>
> [2] Koirala P, Fleming C. Flow-Based Single-Step Completion for Efficient and Expressive Policy Learning[J]. arXiv preprint arXiv:2506.21427, 2025.
>
> [3] Pfrommer S, Huang Y, Sojoudi S. Reinforcement learning for flow-matching policies[J]. arXiv preprint arXiv:2507.15073, 2025.
>
> [4] McAllister D, Ge S, Yi B, et al. Flow matching policy gradients[J]. arXiv preprint arXiv:2507.21053, 2025.
>
> [5] Lipman Y, Chen R T Q, Ben-Hamu H, et al. Flow matching for generative modeling[J]. arXiv preprint arXiv:2210.02747, 2022.
>
> [6] Geng Z, Deng M, Bai X, et al. Mean flows for one-step generative modeling[J]. arXiv preprint arXiv:2505.13447, 2025.

---

> > ### Author Response · Authors · 2025-11-21
> > **Response to Reviewer fH6c (Part 2)**
> >
> > **[Q4]. The current proof seems to largely restate what was in the main text, focused on how one samples and drops constants. Do you believe this is a contraction?**
> >
> > The MeanFlow operator is not guaranteed to satisfy the contraction condition. One can propose some specific design in the average velocity training to enforce the contraction condition. However, the main focus of our paper is proposing a practical online flow policy method with efficient training and sampling, and we find that FPMD-M performs stably in the experiments.
> >
> > **[Q5]. There is a key issue in the experiments that results are reported over 5 seeds. This is insufficient to make strong claims about performance.**
> >
> > Using 3-5 random seeds is the default choice in deep RL papers. For example, among the most classic papers, SAC [7] uses 5 random seeds, and PPO [8] uses 3 random seeds in its experiments. Also, 5 random seeds are used in recent diffusion policy papers such as DACER [9] and DPMD [10].
> > In Table 1 and Table 2, the performance comparison shows a significant advantage over all other NFE=1 algorithms with 5 random seeds, which is sufficient to justify the improvement of our FPMD algorithms.
> >
> > **[Q6] It would also be useful to report training costs to show this tradeoff and better motivate FPMD-M.**
> >
> > Thank you for the suggestion. The training time comparison between our proposed FPMD and diffusion policy method DPMD, DACER is presented in the table below. We conduct the experiments by training for 200K iterations with 1M environment steps on the MuJoCo Ant-v4 environment. We will also include this result in the updated paper.
> >
> > ||DPMD|DACER|FPMD-R|FPMD-M|
> > |---|---|---|---|---|
> > |Training time (min)|21.52 |52.60|21.57|12.53|
> >
> > **[Q7]. Other questions**
> >
> > Thank you for the thoughtful feedback. We will include the training cost comparison and the training curves for state-based environments in the paper. We have fixed the mentioned minor issues and included the recommended citations.
> >
> > [7] Haarnoja T, Zhou A, Abbeel P, et al. Soft actor-critic: Off-policy maximum entropy deep reinforcement learning with a stochastic actor[C]//International conference on machine learning. Pmlr, 2018: 1861-1870.
> >
> > [8] Schulman J, Wolski F, Dhariwal P, et al. Proximal policy optimization algorithms[J]. arXiv preprint arXiv:1707.06347, 2017.
> >
> > [9] Wang Y, Wang L, Jiang Y, et al. Diffusion actor-critic with entropy regulator[J]. Advances in Neural Information Processing Systems, 2024, 37: 54183-54204.
> >
> > [10] Ma H, Chen T, Wang K, et al. Efficient Online Reinforcement Learning for Diffusion Policy[J]. arXiv preprint arXiv:2502.00361, 2025.

---

### Meta-Review · Area_Chair_uzyM · 2026-01-04

**Summary:**

The paper proposes Flow Policy Mirror Descent (FPMD), a method designed to enable efficient one-step sampling for flow-based policies in RL. The authors introduce two variants, FPMD-R (Flow) and FPMD-M (MeanFlow), aiming to balance the expressiveness of diffusion policies with the inference speed of Gaussian policies.

While the motivation to reduce inference latency is well-founded, the paper is recommended for rejection due to and unconvincing validity empirical results identified by the reviewers:
1. A primary concern raised by the reviewers is the lack of convergence guarantees for the MeanFlow operator (Eq. 10). The authors rely on a fixed-point iteration argument but fail to prove the necessary contraction mapping property. During the discussion, this property does not hold generally, weakening the validity of the FPMD-M variant. Additionally, the provided error bounds rely on the assumption of low target variance, which may not hold in complex tasks.

2. The reviewers requested a comparison against state-of-the-art Maximum Entropy diffusion baselines (SDAC). The authors provided this data in the rebuttal, but the results showed that FPMD-R underperforms SDAC on several standard benchmarks, which contradicts the paper's premise of maintaining high performance while improving efficiency.

3. The reviewers noted that the authors utilized the Mish activation function for their method while comparing against baselines that typically use ReLU. This hyperparameter discrepancy casts doubt on whether the reported performance is due to the proposed algorithm or simply better network architecture tuning.

Given the combination of the mentioned weaknesses, the paper does not meet the bar for acceptance at ICLR.

**Reviewer Concerns:**

Addressed Concerns:
1. The reviewers requested a comparison against state-of-the-art Maximum Entropy diffusion policies like SDAC. The authors addressed this by providing a new comparison table in the rebuttal. However, while the request for data was met, the results showed that the proposed FPMD-R method underperformed SDAC on key tasks (e.g., HalfCheetah), weakening the paper's claims of superiority.
2. The reviewer asked for an intuitive explanation of Equation 10. The authors addressed this by framing it as a "fixed-point iteration".

Outstanding Concerns:
1. The most critical outstanding concern is about the validity of the MeanFlow operator (Eq. 10). The authors claim it converges via fixed-point iteration, but did not prove that the operator is a contraction mapping. Without this property, there is no guarantee of convergence, rendering the invalidity of FPMD-M.
2. The authors used the Mish activation function for their method but compared against baselines that likely used ReLU. This creates an unfair experimental setup, as activation functions can significantly impact RL performance. This concern remains unaddressed as the baselines were not retrained with consistent hyperparameters to prove the method's independent value.
3. The theoretical bound for one-step sampling error relies on the assumption that the target distribution variance is small, but the paper does not theoretically prove why or when this variance becomes small during training.
4. The reliance on Importance Sampling (IS) for the flow matching objective introduces high variance, particularly in high-dimensional action spaces. The authors' reliance on heuristic Q-value normalization does not fully resolve the underlying instability of IS weights.

**Reviewer Scores:**

- Reviewer qrLo: unlikely to increase their score.

- Reviewer fH6c: remain negative

- Reviewer onHY: remain negative

- Reviewer GQsJ: remain negative

---

### Decision · Program_Chairs · 2026-01-26

Reject